# LANGUAGE MODELS AS NOISY EXPERTS FOR SEQUENTIAL CAUSAL DISCOVERY

## ABSTRACT

Causal discovery from observational data typically assumes access to complete data and availability of domain experts. In practice, data often arrive in batches, are subject to sampling bias, and expert knowledge is scarce. Language Models (LMs) offer a surrogate for expert knowledge but suffer from hallucinations, inconsistencies, and bias. We present a hybrid framework that bridges these gaps by adaptively integrating sequential batch data with LM-derived noisy, expert knowledge while accounting for both *data-induced* and *LM-induced* biases. We propose a representation shift from Directed Acyclic Graph (DAG) to Partial Ancestral Graph (PAG), that accommodates ambiguities within a coherent framework, allowing grounding the *global* LM knowledge in *local* observational data. To guide LM interactions, we use a sequential optimization scheme that adaptively queries the most informative edges. Across varied datasets, we outperform prior work in structural accuracy and extend to parameter estimation, showing robustness to LM noise.

## 1 INTRODUCTION

Inference of causal relations from observational data remains a challenge in applications across healthcare, economics, business, and scientific discovery (Sanchez et al., 2022; Tu et al., 2019; Sadeghi et al., 2023; Ebert-Uphoff & Deng, 2012). The challenge is addressed through a dual approach: applying causal learning algorithms to observational data while incorporating domain expertise to resolve structural uncertainties (Spirtes et al., 2000; Neapolitan et al., 2004; Spirtes & Zhang, 2016; Chickering, 2002). However, domain expertise can be a scarce resource (He & Geng, 2008; Choo & Shiragur, 2023; Mooij et al., 2016; Meek, 2013; Constantinou et al., 2023). Advanced Language Models (LMs) create opportunities to explore their potential as surrogate experts for causal discovery (Kiciman et al., 2024; Willig et al., 2022). LMs generate informative priors (Takayama et al., 2025) or constraints (Long et al., 2022; Ban et al., 2023b), improving accuracy when combined with data-driven algorithms. Yet, LMs pose their own challenges: hallucination, inconsistency, or failure to capture context-specific nuances (Ji et al., 2023; Kiciman et al., 2024).

The challenges compound since in common applications, observational data arrive batch-wise at a cadence, instead of as a complete dataset. Examples include web and app metrics of all online firms. Privacy regulations and storage constraints may further restrict data access to a short look-back window. This introduces *data-induced* bias, since a batch may suffer from sampling bias that distorts causal discovery. Separately, use of LMs, including *large* ones, poses two problems: *(i)* As surrogates for domain expertise, LMs introduce an *LM-induced* bias—their responses in terms of informative causal priors are prone to hallucinations, contextual brittleness, and inconsistency (Ji et al., 2023; Kiciman et al., 2024). *(ii)* The *global* knowledge encoded in LMs may not align with domain-specific *local* patterns emblematic of batch-wise data, leading to potentially biased learning.

Inattention to the dual biases—data-induced and LM-induced—is a key gap in current approaches to causal discovery with LMs, which we address. First, we propose a major change in representation shift from a Directed Acyclic Graph (DAG), which LM-augmented causal discovery methods currently use, to a Partial Ancestral Graph (PAG), to accommodate uncertainty in the causal structure arising from the dual biases. Second, we propose a novel Bayesian approach to causal structure discovery, where beliefs over causal structure are updated with new data-batch, while augmenting noisy LM-knowledge as priors In support of PAG, we show that popular methods of pairwise and triplet prompting are *overly-optimistic* (*cf.* Table 1) and generate unreliable causal structure in the form of a DAG.

Table 1: **LMs are overly optimistic:** LM based DAG-Pairwise and DAG-triplet prompting methods achieve high recall with low precision across temperatures on two common causal discovery datasets. This limitation calls for explicitly modeling data and LM biases. SHD=Structural Hamming Distance.

| Dataset | Temp. | Method | GPT-3.5$_{turbo}$ | | | GPT-4o | | |
|---|---|---|---|---|---|---|---|---|
| | | | SHD ↓ | Precision ↑ | Recall ↑ | SHD ↓ | Precision ↑ | Recall ↑ |
| EARTHQUAKE | 0.0 | Pairwise | 3.0±0.0 | 0.57±0.00 | 1.0±0.0 | 2.0±0.0 | 0.67±0.00 | 1.0±0.0 |
| | | Triplet | 2.1±0.3 | 0.66±0.03 | 1.0±0.0 | 4.8±0.4 | 0.46±0.02 | 1.0±0.0 |
| | 0.5 | Pairwise | 2.8±0.6 | 0.59±0.05 | 1.0±0.0 | 2.2±0.4 | 0.65±0.04 | 1.0±0.0 |
| | | Triplet | 2.1±0.3 | 0.66±0.03 | 1.0±0.0 | 4.4±0.5 | 0.48±0.03 | 1.0±0.0 |
| | 1.0 | Pairwise | 3.2±0.6 | 0.56±0.05 | 1.0±0.0 | 1.8±0.4 | 0.69±0.05 | 1.0±0.0 |
| | | Triplet | 2.0±0.6 | 0.67±0.07 | 1.0±0.0 | 6.2±0.8 | 0.39±0.03 | 1.0±0.0 |
| ASIA | 0.0 | Pairwise | 25.2±0.4 | 0.24±0.00 | 1.0±0.0 | 11.2±0.4 | 0.42±0.01 | 1.0±0.0 |
| | | Triplet | 24.5±0.5 | 0.25±0.00 | 1.0±0.0 | 19.0±0.6 | 0.30±0.01 | 1.0±0.0 |
| | 0.5 | Pairwise | 23.4±1.0 | 0.26±0.01 | 1.0±0.0 | 11.6±0.5 | 0.48±0.01 | 1.0±0.0 |
| | | Triplet | 24.0±1.0 | 0.25±0.01 | 1.0±0.0 | 19.2±0.8 | 0.29±0.01 | 1.0±0.0 |
| | 1.0 | Pairwise | 23.2±1.5 | 0.25±0.02 | 0.9±0.1 | 11.8±0.4 | 0.40±0.01 | 1.0±0.0 |
| | | Triplet | 23.3±1.0 | 0.26±0.01 | 1.0±0.0 | 26.2±1.7 | 0.23±0.01 | 1.0±0.0 |

Overly optimistic behavior of the LM experts lead to high recall and low precision.

✗ No *global* causal structure
✗ No grounding to *local* data
✗ Need heuristics to postprocess

We introduce BLANCE (Bayesian LM-Augmented Causal Estimation); see Fig. 1 for an overview of the framework. BLANCE differs from existing methods that either rely solely on access to the complete observational data or treat LMs as primary discovery mechanism. In a novelty, the causal structure discovery itself is Bayesian in that the beliefs about causal structure from data are updated iteratively by information drawn from an LM, as data arrive in batches. That is, we adopt a *data-first* approach where for each batch, a traditional causal discovery algorithm (e.g., FCI, (Spirtes et al., 1995)) constructs an initial PAG, which is then iteratively refined through optimized LM queries that leverage the global knowledge while remaining grounded in observed data. To maximize performance under limited budget, LM interactions are framed as a *sequential optimization* problem, selecting the most important edges to query, while accumulating background knowledge over batches. Moreover, to complete the causal discovery process, the parameter (edge weights) estimation we propose is *also* Bayesian, which incorporates potentially noisy LM priors on latent confounders and causal relationships. Our method uncovers cross-sectional causal relationships within each batch, rather than modeling temporal dependencies. BLANCE applies especially to situations where studying contemporaneous causal relations among metrics is separated from confounding temporal effects.

**Contributions** We summarize the contributions as follows: *(i)* We propose a fundamental representation shift from DAGs to PAGs, in a hybrid setup of batch, observational data and LM as noisy expert, that inherently captures uncertainty in causal structure learning. *(ii)* BLANCE—a Bayesian framework for *causal structure* discovery with sequential batch data treating LMs as noisy experts thus accounting for dual sources of bias. *(iii)* A sequential optimization strategy for selecting maximally informative LM edge queries under fixed LM budget constraints. *(iv)* A Bayesian *parameter estimation* algorithm that robustly integrates noisy LM priors with batched data.

## 2 PROBLEM SETUP: SEQUENTIAL CAUSAL DISCOVERY WITH LMS

Traditional causal discovery methods uncover causal structure by exploiting statistical dependencies in observational data, typically assuming access to the complete dataset, and reliable domain knowledge. In contrast, we focus on the setting of sequential, batch-wise observational data. This setting introduces dual sources of bias: *(i)* potentially biased and limited batched observational data, and *(ii)* noisy LM responses. Below we introduce the notation, assumptions, and problem statement. A detailed discussion of related work and how BLANCE differs is provided in App. A.

**Notation and setup** We define batches of observational data $\mathcal{D} = \{\mathbf{D}_i\}_{i=0}^{N}$, where each $i^{\text{th}}$ batch is a sample from the same underlying *'true'* distribution, $\mathbf{D}_i \sim \mathbb{D}$. Each batch contains same set of observed variables, $\mathbf{V}^O$, with $\mathbf{D}_i \in \mathbb{R}^{n_i \times d}$, where $n_i$ is the number of data points, varies by batch, and $d = |\mathbf{V}^O|$ is the number of observed variables. $\mathbb{R}$ denotes real numbers. Any categorical value for an observed variable is encoded as a numerical value. All notations are succinctly shown in Table A1.

For each batch $\mathbf{D}_i$, a causal graph $\mathcal{G}^{D_i} = (\mathbf{V}^O, \mathbf{E}^{D_i})$ is inferred using a standard causal discovery algorithm, where $\mathcal{G}^{D_i}$ is a PAG with $\mathbf{V}^O$ nodes and $\mathbf{E}^{D_i}$ edges. Since $\mathcal{G}^{D_i}$ represents a PAG, $\mathbf{E}^{D_i} \in \mathbf{E}^{PAG}$. We assume there exists a true causal graph $\mathcal{G} = (\mathbf{V}, \mathbf{E})$, where $\mathbf{V}^O \subseteq \mathbf{V}$ and $\mathbf{V} = \{\mathbf{V}^O, \mathbf{V}^L\}$ represents all the variables, both observed and unobserved latent, and $\mathbf{E}$ represents the true causal relationships.

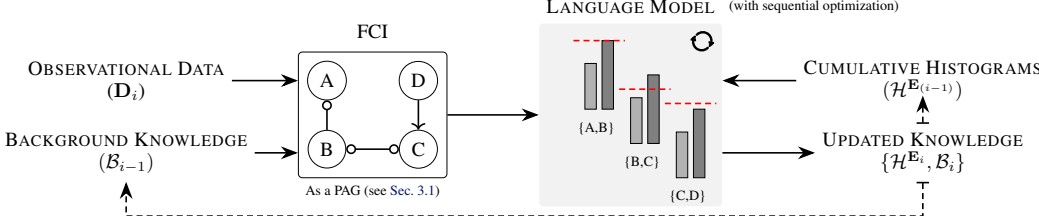

Figure 1: **Overview of the BLANCE framework:** At each batch $\mathbf{D}_i$, observational data is combined with accumulated background knowledge $\mathcal{B}_{(i-1)}$ as prior to estimate a PAG structure via FCI. A language model is then queried—under sequential optimization—to produce beliefs over possible causal relations and update $\mathcal{B}_i$. The updated $\{\mathcal{H}^{\mathbf{E}_i}, \mathcal{B}_i\}$ are fed back into the next iteration.

**Assumptions 1, 2, and 5** are common in prior work and the rest are specific to our setting:

*Assumption 1 [Observability of variables]:* For each $\mathbf{D}_i$, all variables in $\mathbf{V}^{\mathbf{O}}$ are observed across all $n_i$ data points, with $\mathbf{V}^{\mathbf{O}}$ remaining constant across batches. Formally, $[\forall i, \forall v \in \mathbf{V}^{\mathbf{O}}, \forall j \in 1, \ldots, n_i : \mathbf{D}_i[j, v]$ is observed].

*Assumption 2 [Stationary Causal Structure]:* The underlying *'true'* causal structure $\mathcal{G}$ is fixed and does not change across batches.

*Assumption 3 [Selection bias and confounding variables]:* Each batch $\mathbf{D}_i$ may not represent the population accurately due to potential selection bias. Consequently, we *relax* both Faithfulness assumption and Population Inference assumption for each batch. However, we *maintain* the Causal Markov assumption. Additionally, we *relax* the Causal Sufficiency assumption, to allow the presence of latent confounding variables.

*Assumption 4 [Limited look-back window]:* With batches arriving sequentially, only a limited look-back window $k$ of batches is accessible, *i.e.* only the last $k$ batches are available. We also address the challenging data-scarce scenario where $k = 0$, meaning only the current batch is available.

*Assumption 5 [Linear Gaussian model]:* Causal relationships follow a linear relationship between variables $A = \theta_A \times B + \epsilon$, where a directed edge $B \to^{\theta_A} A$ exists and $\epsilon \sim \mathcal{N}(0, \sigma^2)$ represents Gaussian noise whose variance $\sigma^2$ is constant across the structure equation model (SEM).

*Assumption 6 [Single confounder per variable] :* Each variable in $\mathbf{V}^{\mathbf{O}}$ has at most one confounder. Formally, if $A \leftrightarrow B$ exists, then there cannot exist $A \leftrightarrow C$ or $B \leftrightarrow C$ for any other observed variable $C$.

*Assumption 7 [Atomic confounding]:* For $A \leftrightarrow B$, there exists exactly one latent variable, *i.e.* $A \leftarrow L \to B$ is only possible and not $A \leftarrow \{L_1, L_2, \ldots, L_k\} \to B$ or any other combination.

**LM-augmented sequential causal discovery** Traditional hybrid approaches rely on domain experts to narrow the gap between the inferred causal graph $\mathcal{G}^{\mathbf{D}_i}$ and the true causal graph $\mathcal{G}$ by either introducing knowledge about unobserved variables, effectively reducing the set $\mathbf{V} \setminus \mathbf{V}^{\mathbf{O}}$, and adding, removing, reorienting edges in the inferred graph, aiming to minimize the difference $\mathbf{E} \setminus \mathbf{E}^{\mathbf{D}_i}$.

BLANCE's hybrid approach extends the prior art in two directions: *(i)* using an LM as a noisy expert to improve the causal structure $\mathcal{G}^{\mathbf{D}_i}$, obtained via known causal discovery algorithm, where *(ii)* data arrive sequentially in batches. The LM (noisy e**X**pert), represented $X^i$, helps in reducing the Markov equivalence class to yield causal structure ($\mathcal{G}^{\mathbf{X}_i}$). Formally,

$$f_{\mathrm{CD}} : \mathbf{D}_i \to \mathcal{G}^{\mathbf{D}_i} \; ; \; f_{\mathrm{LM}} : \mathcal{G}^{\mathbf{D}_i} \to \mathcal{G}^{\mathbf{X}_i} , \tag{1}$$

where $\mathcal{G}^{\mathbf{D}_i} = (\mathbf{V}^{\mathbf{O}}, \mathbf{E}^{\mathbf{D}_i})$ and $\mathcal{G}^{\mathbf{X}_i} = (\mathbf{V}^{\mathbf{X}_i}, \mathbf{E}^{\mathbf{X}_i})$. The aim is to reduce the discrepancy between the inferred graph and the true causal structure through LM expertise,

$$\mathcal{G} \setminus \mathcal{G}^{\mathbf{X}_i} \leq \mathcal{G} \setminus \mathcal{G}^{\mathbf{D}_i} \; ; \; \mathbf{V} \setminus \mathbf{V}^{\mathbf{X}_i} \leq \mathbf{V} \setminus \mathbf{V}^{\mathbf{O}} \; ; \; \mathbf{E} \setminus \mathbf{E}^{\mathbf{X}_i} \leq \mathbf{E} \setminus \mathbf{E}^{\mathbf{D}_i} . \tag{2}$$

The $\mathbf{V}^{\mathbf{X}_i} \setminus \mathbf{V}^{\mathbf{O}}$ is the set of *LM-suggested* variables while $\mathbf{E}^{\mathbf{X}_i} \setminus \mathbf{E}^{\mathbf{D}_i}$ is the set of *LM-suggested* edges. To integrate the LM's noisy responses and address the inherent bias in batch data, our Bayesian causal discovery framework explicitly handles both *data-induced* and *LM-induced* biases.

**Parameter estimation** Once the LM-augmented causal structure $\mathcal{G}^{\mathbf{X}_i}$ is obtained, we focus on estimating the parameters of the structure equation model (SEM) *i.e.* the edge weights $\boldsymbol{\theta}$ and the noise

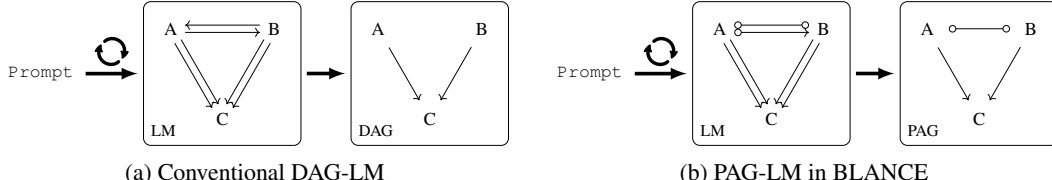

(a) Conventional DAG-LM        (b) PAG-LM in BLANCE

Figure 2: **PAG-LM** streamlines how LMs compose the graph and allows for ambiguities to be indicated in the structure. *(a)* DAG is constructed by iterative prompting (⟳) leading to ambiguities (*e.g.,* ⟷) requiring heuristics that cannot be represented. *(b)* BLANCE represents the causal structure as a PAG that implicitly allow ambiguities to be represented providing a richer representation ( *e.g.,* ○–○)

variance $\sigma^2$; $\phi = \{\boldsymbol{\theta}, \sigma^2\}$. A straightforward method to learn the parameters $\phi$ is Maximum Likelihood Estimation (MLE), $\nabla_\phi \log p(\mathbf{D}_i \mid \mathcal{G}^{X_i}, \phi)$. However, a critical limitation appears in the presence of unobserved *expert-suggested* variables ($\mathbf{V}^L$) in the augmented causal structure, as the likelihood becomes intractable. We address this by proposing a Bayesian parameter estimation algorithm that incorporates the *expert-suggested* information about the unobserved (latent) variable(s), $\mathbf{V}^L$.

## 3   SEQUENTIAL CAUSAL DISCOVERY WITH PAGS

The promise of LMs as proxies for domain expert in causal discovery, typically queried via pairwise or triplet prompts, faces key limitations: *(i)* LMs may provide responses regardless of causal relations among other variables, resulting in inconsistent or cyclic causal graphs; *(ii)* LMs may hallucinate; *(iii)* LMs may be overly optimistic, predicting spurious causal relationships with high recall but low precision (*cf.* Table 1). While prior art addresses *(i)* and *(ii)* through heuristics or auxiliary models to enforce acyclicity and consistency, they add complexity and potentially degrading performance. Crucially, *(ii)* and *(iii)* remain largely unexplored in the setting of sequential batch data where two distinct sources of bias emerge: *LM-induced* bias and *data-induced* bias.

### 3.1   PAG TO INCORPORATE UNCERTAINTY

Prompting methods (pairwise or triplet) constrain the response format to—*'causal', 'non-causal',* or *'unknown'*—which prevent LMs from expressing uncertainty, thus exacerbating bias and inconsistency. To address this, we propose a representational shift from DAG to PAG when using LMs as proxy for experts. PAGs encode uncertainty and structural ambiguity in a principled manner, accommodating latent confounding and partial orientation (*cf.* Fig. 2). Formally, we expand the limited edge set of DAG, $\mathbf{E}^{\text{DAG}} = \{\rightarrow, \leftarrow, \cdot\}$, to the richer set for PAG, $\mathbf{E}^{\text{PAG}} = \{\rightarrow, \leftarrow, \leftrightarrow, \circ\!\!\rightarrow, \leftarrow\!\!\circ, \circ\!\!-\!\!\circ, \cdot, -\}$ [1], where $\cdot$ represents no causal relation. The expansion allows an LM to select from more options and can improve causal discovery (*cf.* Table 2).

Table 2: **DAG to PAG:** Structural Intervention Distance (SID) between DAG-Pairwise (Voting) and PAG-Pairwise depicts benefit of PAG to represent inherent causal uncertainty.

| Dataset | Temp. | Method | SID ↓ |
|---|---|---|---|
| EARTHQUAKE | 0.0 | Pairwise (Voting) | $(1.0, 1.0) \pm (0.0, 0.0)$ |
| | | PAG-Pairwise | $(0.0, 0.0) \pm (0.0, 0.0)$ |
| | 0.5 | Pairwise (Voting) | $(1.4, 1.4) \pm (1.2, 1.2)$ |
| | | PAG-Pairwise | $(0.0, 0.0) \pm (0.0, 0.0)$ |
| | 1.0 | Pairwise (Voting) | $(1.6, 1.6) \pm (1.5, 1.5)$ |
| | | PAG-Pairwise | $(0.8, 0.8) \pm (1.6, 1.6)$ |
| ASIA | 0.0 | Pairwise (Voting) | $(6.4, 6.4) \pm (0.8, 0.8)$ |
| | | PAG-Pairwise | $(3.8, 3.8) \pm (0.9, 0.9)$ |
| | 0.5 | Pairwise (Voting) | $(5.2, 5.2) \pm (1.7, 1.7)$ |
| | | PAG-Pairwise | $(3.5, 3.5) \pm (0.8, 0.8)$ |
| | 1.0 | Pairwise (Voting) | $(5.4, 5.4) \pm (1.8, 1.8)$ |
| | | PAG-Pairwise | $(4.0, 4.0) \pm (1.9, 1.9)$ |

The representational shift to PAG is a necessary pivot and starting point for addressing dual sources of bias—*LM-induced* and *data-induced*. The LM's error prone response constitutes a prior and motivates a novel Bayesian causal structure discovery framework in the sequential batch setting, whereby we integrate causal predictions from observational data and LMs in a sequential, iterative manner. We take the results from Table 2 further and introduce BLANCE—LM-augmented Bayesian framework for causal structure learning, followed by Bayesian parameter estimation.

---

[1]To capture selection bias, ambiguous edges (○–○, ○→) and undirected edges '−' can be used. Not all causal discovery algorithms output undirected edges (*e.g.,* https://causal-learn.readthedocs.io/en/latest/search_methods_index/Constraint-basedcausaldiscoverymethods/FCI.html), but they output ambiguous edges, which we use as proxy for selection bias.

## 3.2 BAYESIAN LM-AUGMENTED CAUSAL STRUCTURE DISCOVERY

As a principled way to incorporate the dual sources of bias, we adopt a Bayesian formulation. Formally, given data batch $\mathbf{D}_i$, cumulative background knowledge $\mathcal{B}_{(i-1)}$ up to batch $(i-1)$, the posterior over $\mathcal{G}_i$ is obtained via Bayes' rules,

$$\underbrace{p(\mathcal{G}_i \mid \mathbf{D}_i)}_{\text{Posterior}} \propto \underbrace{p(\mathbf{D}_i \mid \mathcal{G}_i)}_{\text{Likelihood}} \underbrace{p(\mathcal{G}_i \mid \mathcal{B}_{(i-1)})}_{\text{Prior}}. \tag{3}$$

The prior $p(\mathcal{G}_i \mid \mathcal{B}_{(i-1)})$ is iteratively shaped by background knowledge, which accumulates across batches. Intuitively, the prior encodes the edge type between node pairs in the causal structure. Once the posterior is obtained, an LM is queried to update and obtain background knowledge $\mathcal{B}_i$ using Eq. (5) and Eq. (9), which shapes the prior for the next batch.

Given the inherent stochasticity of LMs, its response can be viewed as a *sample* from an implicit distribution over the edge types ($\mathbf{E}^{\text{PAG}}$). This allows explicit modeling of uncertainty in LM responses, $f_{\text{LM}}^{(i)}(A, B, \text{Pr}) \sim p(E_{AB} \mid A, B)$, where $A, B$ are the two nodes, Pr is the prompt, and $E_{AB} \in \mathbf{E}^{\text{PAG}}$. Treating LM responses as noisy observations rather than ground truth, addresses the challenge of LM hallucination. As more batches are processed, the prompt Pr becomes more informative, thereby decreasing uncertainty in LM responses. Formally, we treat LM as a *black-box causal edge sampler* and aggregate multiple LM samples into empirical histograms that are updated iteratively over batches,

$$\mathcal{H}^{\mathbf{E}_i}(A, B)[E_{AB}] = \mathcal{H}^{\mathbf{E}_{i-1}}(A, B)[E_{AB}] + \mathbb{I}[f_{\text{LM}}^{(i)}(A, B) = E_{AB}], \tag{4}$$

where $\mathcal{H}^{\mathbf{E}_i}(A, B)$ represents the cumulative histogram up to batch $i$, and $E_{AB}$ represents a type of causal relation. These histograms define an approximate posterior distribution over edge types, capturing the LM's evolving beliefs about causal relationships.

To convert these histogram representations into background knowledge $\mathcal{B}_i$, we introduce a dynamic threshold metric for each candidate edge $e$ in the histogram $\mathcal{H}^{\mathbf{E}_i}$,

$$\tau_i^e = \alpha \times E_i^e \times T_i^e + (1 - \alpha)\sqrt{T_i^e\left(1 - \frac{T_i^e}{T_i}\right)}, \quad \text{s.t.} \quad E_i^e = -\sum_j \frac{\mathcal{H}_{j,e}^{\mathbf{E}_i}}{T_i^e} \times \log\left(\frac{\mathcal{H}_{j,e}^{\mathbf{E}_i}}{T_i^e}\right). \tag{5}$$

Here, for batch $i$ and edge $e$, $\tau_i^e$ denotes the threshold, $E_i^e$ the posterior entropy, $T_i^e$ the number of LM interactions, $T_i = \sum_e T_i^e$ the total interactions, and $\mathcal{H}_{j,e}^{\mathbf{E}_i}$ the frequency of bin $j$ in $e$'s histogram.

Intuitively, the first term in $\tau_i^e$ of Eq. (5) accounts for uncertainty in the histogram edge distribution, while the second term handles the sampling uncertainty that decreases as more batches of data arrive. The hyperparameter $\alpha$ balances between these two terms. Fig. 4 showcases the efficiency of the proposed dynamic background threshold (*cf.* Eq. (5)), with the additional details discussed in Sec. 5.1. The pseudo-code of the algorithm is outlined in Alg. 1.

## 3.3 LM INTERACTION: SEQUENTIAL OPTIMIZATION

Given the stochastic LM responses and cumulative histogram-based estimates of edge uncertainty, we next address how to allocate LM queries efficiently across candidate edges. We model LM interactions $f_{\text{LM}}$ as a *sequential optimization* problem under a limited budget. At each batch $i$, up to $m^{\text{L}}$ calls to the LM are allowed. The objective is to strategically allocate these calls to refine the edge distribution $\mathcal{H}^{\mathbf{E}_i}$ and expand the set of background knowledge $\mathcal{B}_i$.

In the edge refinement setting, each query corresponds to selecting a candidate edge $e$ and querying $f_{\text{LM}}$ to reduce uncertainty about its type. The LM's response is treated as a noisy sample from the underlying distribution over edge types. This induces a natural trade-off: we must *explore* uncertain edges to improve estimates and *exploit* promising edges that are likely to yield useful and increasing background knowledge.

---

**Algorithm 1** LM-augmented structure learning

**Require:** $\mathbf{D}_i, \mathcal{H}^{\mathbf{E}(i-1)}, \mathcal{H}^{\text{L}(i-1)}, \mathcal{B}_{(i-1)}, m^{\mathbf{E}}, m^{\text{L}}$
**Ensure:** $\mathcal{H}^{\mathbf{E}_i}, \mathcal{H}^{\text{L}_i}, \mathcal{B}_i$
1: **Initial causal structure**
2: $f_{\text{CD}} : \mathbf{D}_i \times \mathcal{B}_{(i-1)} \to \mathcal{G}^{\text{D}_i}$
3: **Expert-guided causal structure refinement**
4: $f_{\text{LM}} : \mathcal{G}^{\text{D}_i} \times \mathcal{H}^{\mathbf{E}(i-1)} \times \mathcal{B}_{(i-1)} \times m^{\mathbf{E}} \to (\mathcal{H}^{\mathbf{E}_i}, \mathcal{B}_i)$
5: **Expert-suggested latent confounder**
6: **for** $A \leftrightarrow B$ in $\mathcal{B}_i$ **do**
7: $\quad f_{\text{LM}} : \mathcal{H}^{\mathbf{E}_i} \times \mathcal{H}^{\text{L}(i-1)} \times A \times B \times m^{\text{L}} \to \mathcal{H}^{\text{L}_i}$
8: **end for**
9: **return** $\mathcal{H}^{\mathbf{E}_i}, \mathcal{H}^{\text{L}_i}, \mathcal{B}_i$

---

Table 3: **BLANCE improves causal discovery:** We experiment with six datasets–number of observed variables ranges 5 to 37– using two paradigms: *Only-Data* and *Data-LM*. We evaluate with 5 metrics: *Modified SHD, SID, Precision, Recall, F1*. All methods use GPT-3.5$_{\text{turbo}}$ as an LM with temperature 1. We report mean and standard deviation over 5 runs and perform significance test with $\alpha = 0.05$.

| Dataset | Approach | Method | Mod. SHD ↓ | SID ↓ | Precision ↑ | Recall ↑ | F1 Score ↑ |
|---|---|---|---|---|---|---|---|
| EARTHQUAKE ($d=5$) | Only-Data | FCI-Cumulative | 2.00±0.00 | (0.00, 5.00)±(0.00, 0.00) | **1.00**±0.00 | 0.50±0.00 | 0.67±0.00 |
| | | FCI-Vanilla | 3.60±0.80 | (8.20, 9.20)±(3.60, 1.60) | 0.20±0.40 | 0.05±0.10 | 0.08±0.16 |
| | | FCI-Iterative | 5.00±1.67 | (12.20, 12.20)±(4.66, 4.66) | 0.30±0.27 | 0.20±0.19 | 0.24±0.22 |
| | | FCI-Heuristics | 3.60±0.80 | (8.20, 9.20)±(3.60, 1.60) | 0.20±0.40 | 0.05±0.10 | 0.08±0.16 |
| | Data-LM | LLM-first | 6.00±0.82 | (15.00, 15.00)±(0.82, 0.82) | 0.11±0.16 | 0.08±0.12 | 0.09±0.13 |
| | | ILS-CSL | 2.38±0.96 | (5.25, 6.50)±(0.83, 2.69) | **0.88**±0.22 | 0.44±0.21 | 0.56±0.21 |
| | | BLANCE | **1.00**±0.63 | **(2.20, 2.20)**±(1.60, 1.60) | **1.00**±0.00 | **0.75**±0.16 | **0.85**±0.11 |
| ASIA ($d=8$) | Only-Data | FCI-Cumulative | 7.00±0.00 | (23.00, 49.00)±(0.00, 0.00) | 0.00±0.00 | 0.00±0.00 | 0.00±0.00 |
| | | FCI-Vanilla | 7.80±0.75 | (30.00, 35.00)±(5.90, 2.45) | 0.00±0.00 | 0.00±0.00 | 0.00±0.00 |
| | | FCI-Iterative | 8.00±1.26 | (33.00, 33.00)±(7.46, 7.46) | 0.45±0.24 | 0.23±0.15 | 0.29±0.18 |
| | | FCI-Heuristics | 7.80±0.75 | (30.00, 35.00)±(5.90, 2.45) | 0.00±0.00 | 0.00±0.00 | 0.00±0.00 |
| | Data-LM | LLM-first | 7.33±0.94 | (27.67, 27.67)±(2.49, 2.49) | 0.58±0.12 | 0.29±0.06 | 0.39±0.08 |
| | | ILS-CSL | 6.50±0.50 | (28.50, 28.50)±(3.20, 3.20) | **0.79**±0.12 | 0.28±0.11 | 0.40±0.12 |
| | | BLANCE | **4.60**±1.02 | **(13.60, 13.60)**±(3.83, 3.83) | **0.80**±0.12 | **0.60**±0.12 | **0.67**±0.08 |
| USER LEVEL DATA-I ($d=9$) | Only-Data | FCI-Cumulative | 15.00±2.77 | (47.80, 47.80)±(4.62, 4.62) | 0.72±0.18 | 0.37±0.09 | 0.47±0.09 |
| | | FCI-Vanilla | 21.30±3.50 | (61.60, 61.60)±(5.54, 5.54) | 0.41±0.18 | 0.22±0.10 | 0.29±0.13 |
| | | FCI-Iterative | **5.60**±1.20 | (23.40, 23.40)±(7.94, 7.94) | **0.93**±0.04 | 0.77±0.02 | **0.84**±0.03 |
| | | FCI-Heuristics | 21.30±3.50 | (61.60, 61.60)±(5.54, 5.54) | 0.41±0.18 | 0.22±0.10 | 0.29±0.13 |
| | Data-LM | LLM-first | 9.68±1.67 | (38.12, 38.12)±(4.06, 4.06) | 0.80±0.06 | 0.66±0.04 | 0.72±0.05 |
| | | ILS-CSL | 9.20±1.72 | (34.20, 34.20)±(3.49, 3.49) | 0.82±0.05 | 0.66±0.05 | 0.73±0.05 |
| | | BLANCE | 5.00±0.71 | **(13.75, 13.75)**±(2.49, 2.49) | 0.90±0.04 | **0.83**±0.02 | **0.86**±0.02 |
| USER LEVEL DATA-II ($d=8$) | Only-Data | FCI-Cumulative | 19.80±2.04 | **(40.00, 40.00)**±(3.03, 3.03) | 0.15±0.07 | 0.10±0.04 | 0.12±0.05 |
| | | FCI-Vanilla | 17.40±1.83 | **(36.80, 39.40)**±(2.04, 3.38) | 0.07±0.13 | 0.02±0.03 | 0.03±0.05 |
| | | FCI-Iterative | 20.60±1.77 | (42.80, 43.40)±(4.07, 4.22) | 0.06±0.08 | 0.05±0.07 | 0.05±0.07 |
| | | FCI-Heuristics | 17.40±1.83 | **(36.80, 39.40)**±(2.04, 3.38) | 0.07±0.13 | 0.02±0.03 | 0.03±0.05 |
| | Data-LM | LLM-first | 18.75±0.43 | (44.00, 44.00)±(1.00, 1.00) | 0.18±0.01 | **0.17**±0.00 | **0.18**±0.00 |
| | | ILS-CSL | **16.90**±1.50 | (44.40, 47.80)±(6.05, 3.49) | 0.16±0.03 | 0.10±0.04 | 0.12±0.03 |
| | | BLANCE | **16.33**±1.80 | (40.17, 40.17)±(3.14, 3.14) | **0.26**±0.04 | 0.18±0.06 | **0.20**±0.04 |
| CHILD ($d=19$) | Only-Data | FCI-Cumulative | 27.50±0.00 | **(111.00, 131.00)**±(0.00, 0.00) | 0.38±0.00 | 0.36±0.00 | 0.37±0.00 |
| | | FCI-Vanilla | 28.00±1.48 | (129.20, 133.20)±(10.46, 10.76) | 0.38±0.04 | 0.26±0.05 | 0.31±0.04 |
| | | FCI-Iterative | 32.10±1.16 | (149.00, 164.40)±(7.16, 10.33) | 0.27±0.03 | 0.26±0.04 | 0.26±0.03 |
| | | FCI-Heuristics | 28.00±1.48 | (129.20, 133.20)±(10.46, 10.76) | 0.38±0.04 | 0.26±0.05 | 0.31±0.04 |
| | Data - LLM | LLM-first | 31.67±2.05 | (172.00, 172.00)±(13.42, 13.42) | 0.29±0.05 | 0.30±0.05 | 0.30±0.05 |
| | | ILS-CSL | 32.00±2.10 | (154.00, 154.00)±(26.53, 26.53) | 0.28±0.05 | 0.28±0.05 | 0.28±0.05 |
| | | BLANCE | **25.50**±0.89 | (103.40, 112.20)±(8.09, 7.05) | **0.43**±0.01 | **0.44**±0.02 | **0.43**±0.01 |
| ALARM ($d=37$) | Only-Data | FCI-Cumulative | **45.00**±0.00 | **(626.00, 626.00)**±(0.00, 0.00) | 0.25±0.00 | 0.02±0.00 | 0.04±0.00 |
| | | FCI-Vanilla | 49.50±1.61 | (617.80, 699.20)±(29.23, 68.43) | 0.00±0.00 | 0.00±0.00 | 0.00±0.00 |
| | | FCI-Iterative | 52.40±6.21 | **(612.20, 636.80)**±(49.87, 43.83) | **0.33**±0.14 | 0.12±0.05 | 0.17±0.07 |
| | | FCI-Heuristics | 49.50±1.61 | (617.80, 699.20)±(29.23, 68.43) | 0.00±0.00 | 0.00±0.00 | 0.00±0.00 |
| | Data - LLM | LLM-first | 52.33±3.09 | (673.33, 673.33)±(25.63, 25.63) | **0.33**±0.08 | 0.13±0.02 | 0.19±0.03 |
| | | ILS-CSL | 51.20±1.60 | (657.20, 657.20)±(24.07, 24.07) | 0.34±0.05 | 0.10±0.01 | 0.15±0.02 |
| | | BLANCE | 50.90±1.32 | **(589.80, 591.00)**±(26.48, 25.59) | **0.42**±0.03 | **0.22**±0.02 | **0.29**±0.02 |

Formally, we cast LM interactions as a sequential decision-making problem:

$$\textbf{Arms:} \quad \mathcal{A} = \{\text{All possible edges between variables, } \mathbf{E}^{\text{PAG}}\}, \tag{6}$$

$$\textbf{Reward:} \quad r_k(e) = \text{Information gain from querying edge } e \text{ at step } k, \tag{7}$$

$$\textbf{Policy:} \quad \pi : \mathcal{H}^{\mathbf{E}_i} \times \mathcal{G}^{\text{D}_i} \to e \quad \text{(Edge selection rule).} \tag{8}$$

The optimization objective is to maximize cumulative information gain over $m^{\text{L}}$ LM calls, balancing both the expansion of background knowledge, and the uncertainty reduction in $\mathcal{H}^{\mathbf{E}_i}$. To implement $\pi$, we propose a scoring function that jointly accounts for epistemic uncertainty, proximity to background knowledge thresholds, and exploration,

$$S_i^e = w_1 E_i^e + w_2 \left( \frac{1}{TD_i^e} \right) + w_3 \sqrt{\frac{\log T_i}{T_i^e}}, \quad \text{s.t.} \quad TD_i^e = \tau_i - \max(\mathcal{H}^{\mathbf{E}_i}(e)), \tag{9}$$

where $TD_i^e$ is the threshold distance from being included in background knowledge, $E_i^e, T_i, T_i^e$ are as defined in Eq. (5), and $w_1, w_2, w_3$ are hyper-parameters controlling the trade-off. At each step, the edge $e^* = \arg \max_e S_i^e$ is selected for LM interaction. Fig. 4 showcases the effectiveness of the proposed selection score (*cf.* Eq. (9)), with details discussed in Sec. 5.1.

This formulation generalizes to other expert-guided tasks (*e.g.,* confounder detection) by redefining the arms and reward function. We discuss the connection of the proposed sequential optimization with multi-arm bandits in App. E.

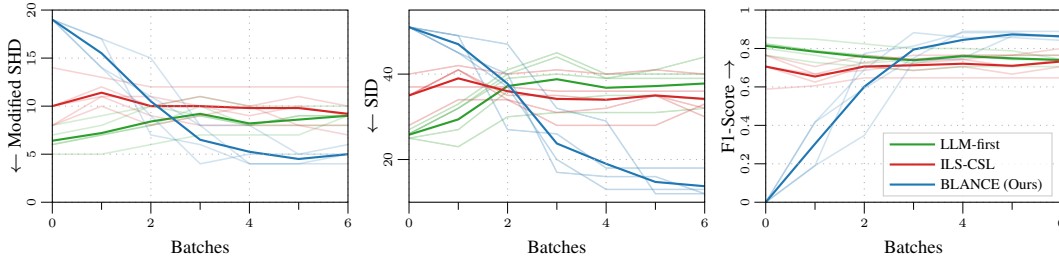

Figure 3: USER LEVEL DATA - I: Performance evolution across batches for *Data-LM* methods. Left: Modified Structural Hamming Distance (↓), Middle: Structural Intervention Distance (↓), and Right: F1-Score (↑). BLANCE consistently outperforms other approaches as data accumulation progresses.

# 4 BAYESIAN PARAMETER ESTIMATION

Once we obtain a causal structure $\mathcal{G}^{X_i}$, we address the critical task of *parameter estimation* within the Structural Equation Model (SEM). The parameters $\phi = \{\theta, \sigma^2\}$ include edge weights (coefficients) and noise parameters. With $\mathcal{G}^{X_i}$ potentially containing both observed and latent variables $\mathbf{V}^{X_i} = \{\mathbf{V}^O, \mathbf{V}^L\}$, we represent observed variable edges as $\theta^O$ and latent confounder edges as $\theta^L$, giving $\theta = \{\theta^O, \theta^L\}$.

When no latent confounders exist ($\mathbf{V}^L = \emptyset$), standard Maximum Likelihood Estimation (MLE) optimizes the parameters $\phi = \{\theta^O, \sigma^2\}$ as $\phi = \arg\max_\phi \log p(\mathbf{D}_i \mid \mathcal{G}^{X_i}, \phi)$ using a conventional gradient-based methods. We focus on the more important and challenging scenario where latent confounders exist ($\mathbf{V}^L \neq \emptyset$).

**Algorithm 2** Bayesian Parameter Estimation

**Require:** $\mathbf{D}_i, \eta, \mathcal{G}^{X_i}, \mathcal{I}^P, \mathcal{I}^\rho$
**Ensure:** $\phi$
1: **Initialization**
2: Warm-start for $\phi^O$
$\qquad \phi^O \in \arg\max_{\phi^O} p(\mathbf{D}_i \mid \mathcal{G}^{X_i}_{-\mathbf{V}^L}, \phi^O)$
3: Get prior over $\mathbf{V}^L$
$\qquad f_{LM} : \mathcal{G}^{X_i} \times \mathbf{V}^L \times \mathcal{I}^P \to \mathcal{N}(\boldsymbol{m}_p, \boldsymbol{S}_p)$
4: Get correlation $\rho(\mathbf{V}^L, \mathbf{V}^O)$
$\qquad f_{LM} : \mathcal{G}^{X_i} \times \mathcal{I}^\rho \to \rho(\mathbf{V}^L, \mathbf{V}^O)$
5: Initialize $\theta^L$ to $\rho(\mathbf{V}^L, \mathbf{V}^O)$ or randomly
6: **Iterative Optimization**
7: **while** not converged **do**
8: $\quad$ Compute posterior over $\mathbf{V}^L$ using Eq. (10)
9: $\quad$ Optimize $\phi$ variables using Eq. (11)
10: **end while**
11: **return** $\phi$

With latent confounders, MLE is ill-posed and intractable. We instead employ an iterative Expectation–Maximization (EM) algorithm that incorporates LM-provided probability $p(\mathbf{V}^L)$ and correlation $\rho(\mathbf{V}^O, \mathbf{V}^L)$ about latent confounders. Specifically, we propose the following EM steps:

- **E-step:** Compute conditional posterior of latent confounder(s) given $\mathbf{D}_i$ and SEM parameters $\phi$,

$$p(\mathbf{V}^L \mid \mathcal{G}^{X_i}, \mathbf{D}_i, \phi) \propto p(\mathbf{D}_i \mid \mathcal{G}^{X_i}, \mathbf{V}^L, \phi) p(\mathbf{V}^L). \tag{10}$$

- **M-step:** Update parameters by maximizing the expected log-likelihood, incorporating LM-provided regularization for latent confounder edges,

$$\phi \in \arg\max_\phi \mathbf{E}_{p(\mathbf{V}^L \mid \mathcal{G}^{X_i}, \mathbf{D}_i, \phi)}[\log p(\mathbf{D}_i \mid \mathcal{G}^{X_i}, \mathbf{V}^L, \phi)] - \lambda \|\theta^L - (\rho(\mathbf{V}^O, \mathbf{V}^L)\sigma_{\mathbf{V}^O}\sigma_{\mathbf{V}^L}^{-1})\|_2. \tag{11}$$

Alg. 2 details the EM parameter estimation algorithm. In Sec. 5.2, we demonstrate the robustness and recovery capability of the proposed parameter estimation algorithm.

# 5 EXPERIMENTS

To provide a strong empirical examination, we conduct experiments on six datasets: EARTHQUAKE (Korb & Nicholson, 2010), ASIA (Lauritzen & Spiegelhalter, 1988), USER LEVEL DATA - I, USER LEVEL DATA - II (Google & Kaggle, 2018), CHILD (Spiegelhalter et al., 1993), and ALARM (Beinlich et al., 1989). They range in number of observed variables from small (5) to medium (19) to large (37). EARTHQUAKE, ASIA, CHILD, and ALARM datasets are standard benchmarks, providing observational data and ground-truth causal structures. To simulate a streaming batch setting, each dataset is split into batches.

For the two USER LEVEL DATA, which contain only observational data, the underlying DAG is inferred using DirectLiNGAM (Shimizu et al., 2011). Each batch is then independently simulated

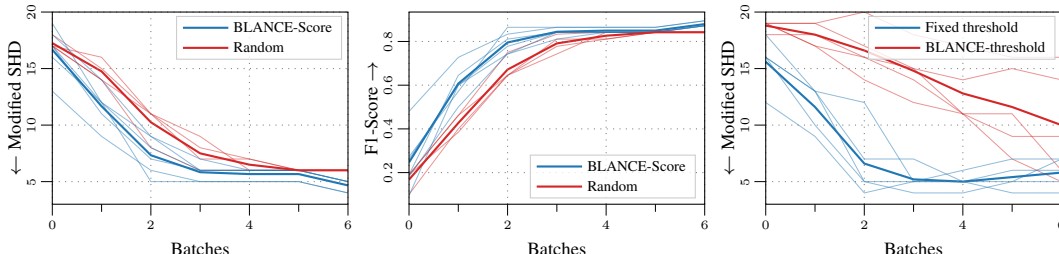

Figure 4: **Structure learning ablation:** The impact of two key components: *selection score* and *dynamic background threshold*. (Left, Middle) Modified SHD (↓) and F1-score (↑) on the USER LEVEL DATA - I dataset, comparing BLANCE—*selection score* against random selection. (Right) Modified SHD(↓) comparing BLANCE— *dynamic threshold* with a conventional fixed threshold.

from the DAG using an SEM, consistent with Assumption 5. We treat the DAG inferred from DirectLiNGAM as the ground truth for evaluation. Further details on the data sets and simulation process are provided in App. B.

Our evaluation metrics include a *modified* Structural Hamming Distance (Mod. SHD), which extends SHD to account for uncertain edges in PAG; Structural Intervention Distance (SID); and precision, recall, and F1-score for causal relations that are certain. Together, these metrics assess structural accuracy, interventional soundness, and edge-wise discovery performance (details in App. D).

## 5.1 STRUCTURE LEARNING

We evaluate BLANCE against *Only-Data* and *Data-LM* baselines in Table 3. As shown in Table 1, *Only-LM* methods exhibit *overly optimistic* behavior, producing globally plausible but locally unreliable causal structures. This highlights the need for data-grounded post-processing. Table 3 compares BLANCE with several baselines, including multiple FCI variants (cumulative, vanilla, iterative, heuristics), as well as *Data-LM* approaches (LM-first, and ILS-CSL (Ban et al., 2023a)). Across all evaluation metrics, BLANCE consistently outperforms baselines, which also holds for different LM temperatures (Table A3). While Table 3 reports metrics for the final batch, we also show performance evolution across batches, a crucial step in sequential settings (*cf.* Fig. 3). We justify the use of FCI over other causal discovery algorithms in App. C. We provide more experiment details in App. G.

Finally, going beyond GPT-3.5$_{turbo}$ (Table 3), results with recent LMs, GPT-4o and GPT-5 (Table A2) show good performance gains for BLANCE across LMs. The much higher inference cost of GPT-4o and GPT-5, over GPT-3.5$_{turbo}$ constrain their use for large set of experiments.

**Structure learning ablations** We ablate two key components of BLANCE: *(i)* selection score for sequential optimization, and *(ii)* dynamic background threshold. Results of ablations performed on USER LEVEL DATA - I are shown in Fig. 4.

Effectiveness of proposed selection score (Eq. (9)) in guiding edge selection under a fixed budget of LM queries, is compared against a random-selection baseline. Fig. 4 (Left, Middle) shows that BLANCE achieves significantly better Mod. SHD and F1-score across batches, demonstrating the benefit of a principled edge selection policy in sequential structure learning.

We also assess the impact of dynamic background threshold (Eq. (5)) used to promote edges from the histogram $\mathcal{H}^{\mathbf{E}}$ into the background knowledge $\mathcal{B}$. Fig. 4 (Right) reports Mod. SHD over batches, highlighting the advantages of a dynamic threshold over a conventional fixed one. The adaptive mechanism yields more stable and accurate graph recovery throughout the learning process.

## 5.2 PARAMETER ESTIMATION

The datasets used in Sec. 5.1 do not contain latent confounders in their causal structures. Consequently, parameter estimation reduces to standard maximum likelihood estimation, which BLANCE (Sec. 4) replicates by design. To meaningfully evaluate BLANCE in the presence of latent confounders, we consider a real-world dataset: the RED-WINE QUALITY (Cortez et al., 2009) (details in App. B).

Table 4: **LM-predicted priors for confounding variable:** LM suggest relevant Gaussian priors when the confounder is meaningful and default to $\mathcal{N}(0,1)$ when uncertain.

| Variables | GPT-3.5$_{turbo}$ | GPT-4o |
|---|---|---|
| density ← alcohol → quality | $\mathcal{N}(12.5, 2.5)$ | $\mathcal{N}(10.5, 1.2)$ |
| density ← alcohol → volatile-acidity | $\mathcal{N}(12.5, 2.5)$ | $\mathcal{N}(10.5, 1.2)$ |
| cat ← alcohol → mouse | $\mathcal{N}(0,1)$ | $\mathcal{N}(0,1)$ |
| bed ← alcohol → shopping | $\mathcal{N}(0,1)$ | $\mathcal{N}(0,1)$ |

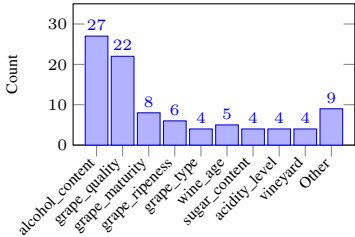

Figure 5: LM predicted confounders.

For the RED-WINE QUALITY, BLANCE's structure learning algorithm predicts a latent confounder between variables *quality* and *density*. Applying DirectLiNGAM on the full dataset indicates that the true confounder is *alcohol content*, aligning with domain knowledge. Following Sec. 3.2, we query an LM to identify potential latent confounders using world knowledge. Fig. 5 shows a histogram of LM predictions, with *alcohol_content* emerging as the top candidate. We incorporate this LM-provided confounder into BLANCE's parameter estimation pipeline and query the LM for its conditional distribution. Table 4 presents the LM-provided conditional Gaussian distributions. Notably, when queried with obscure variables (*e.g.,* (cat, mouse)), LM often defaults to unit Gaussian $\mathcal{N}(0,1)$, akin to using an uninformative prior in Bayesian inference.

To assess parameter recovery, we track evolution of $\boldsymbol{\theta}$ over sequential batches. As a performance metric, we compute the $\ell_2$-norm error $\|\boldsymbol{\theta}^{\star} - \boldsymbol{\theta}\|_2$, where $\boldsymbol{\theta}^{\star}$ denotes the parameters obtained via MLE assuming the confounder (*alcohol_content*) is observed. Fig. 6 visualizes this convergence behavior.

**Parameter estimation: robustness and recovery** We demonstrate robustness of the proposed parameter estimation algorithm under misspecified or ill-informed priors. Based on domain knowledge and observational data, we estimate latent confounder *alcohol_content* to follow distribution $\mathcal{N}(11, 1.0)$. To stress-test BLANCE, we experiment with three alternative priors: $\mathcal{N}(12.5, 2.5)$ (suggested by GPT-3.5$_{turbo}$), $\mathcal{N}(0,1)$, and a severely misspecified prior $\mathcal{N}(50, 1.5)$. Fig. 6 illustrates the evolution of the learned parameters $\boldsymbol{\theta}$ across training batches for each prior. As expected and aligned with the Bayesian principle, convergence is slower when initialized with an inaccurate prior. Nevertheless, the model progressively refines its estimates as more data is processed, ultimately converging towards $\boldsymbol{\theta}^{\star}$. This demonstrates both the robustness and recovery capabilities of BLANCE's Bayesian parameter estimation algorithm—even under poor initialization. Additionally, we incorporate LM-suggested Pearson correlation coefficients in the *M-step* objective to further guide estimation, Eq. (11) (see discussion and results in App. F.).

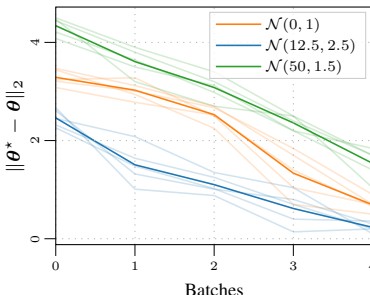

Figure 6: **Parameter Estimation:** Convergence of parameters and robustness to prior misspecification as more batches are processed.

### 5.3 OTHER LM FAMILIES AND MEMORIZATION IN LM

We evaluate BLANCE using three *recent* LMs across different families: Llama3.1$_{8B\text{-}instruct}$, Qwen3$_{4B\text{-}instruct}$, and GPT4.1$_{nano}$. The first two are open models. We evaluate on two datasets: CHILD and the real-world USER LEVEL DATA - I (Google & Kaggle, 2018). Measuring performance on the 5 metrics over 5 runs, the results in Table 5 are consistent with those of the previously presented GPT3.5$_{Turbo}$ model, demonstrating robustness across LMs with different training data, and architectures.

We introduce an additional baseline which does causal discovery using LM and Pearson correlation in the prompt Jiralerspong et al. (2024b) (hereafter BFS). A comparison of BLANCE with BFS also serves an important purpose by throwing light on the issue of potential memorization of datasets seen in training by LMs. As noted by Jiralerspong et al. (2024b), their method relies on the LM's training knowledge (Sec. 5 of their paper), thereby recognizing the dependence on memorization. To make a fair comparison, we use a recent model GPT4.1$_{nano}$. Results reported in the bottom panel of Table 5

Table 5: **BLANCE improves causal discovery over language models:** We showcase results on the CHILD and USER LEVEL DATA-I datasets using three language models: *Llama3.1$_{8B\text{-}instruct}$*, *Qwen3$_{4B\text{-}instruct}$* and a GPT-4.1$_{nano}$. We report 5 metrics: *Modified SHD, SID, Precision, Recall, F1*, with mean and standard deviation over 5 runs and perform significance test with $\alpha = 0.05$.

| Dataset | Model | Method | Mod. SHD ↓ | SID ↓ | Precision ↑ | Recall ↑ | F1 Score ↑ |
|---|---|---|---|---|---|---|---|
| CHILD | Llama3.1$_{8B\text{-}instruct}$ | LLM-first | $45.80\pm1.94$ | $(211.20, 211.20)\pm(15.25, 15.25)$ | $0.13\pm0.01$ | $0.19\pm0.02$ | $0.16\pm0.01$ |
| | | ILS-CSL | $27.60\pm2.24$ | $(146.40, 146.40)\pm(15.93, 15.93)$ | $0.37\pm0.05$ | $0.38\pm0.07$ | $0.38\pm0.06$ |
| | | BLANCE | $\mathbf{24.60}\pm0.80$ | $\mathbf{(101.80, 101.80)}\pm(8.11, 8.11)$ | $\mathbf{0.44}\pm0.01$ | $\mathbf{0.45}\pm0.02$ | $\mathbf{0.44}\pm0.01$ |
| | Qwen3$_{4B\text{-}instruct}$ | LLM-first | $35.80\pm2.04$ | $(173.80, 173.80)\pm(14.13, 14.13)$ | $0.27\pm0.02$ | $\mathbf{0.42}\pm0.02$ | $0.34\pm0.01$ |
| | | ILS-CSL | $30.60\pm2.42$ | $(196.80, 196.80)\pm(14.13, 14.13)$ | $0.30\pm0.06$ | $0.28\pm0.05$ | $0.29\pm0.05$ |
| | | BLANCE | $\mathbf{21.20}\pm0.75$ | $\mathbf{(107.80, 107.80)}\pm(13.42, 13.42)$ | $\mathbf{0.51}\pm0.01$ | $0.32\pm0.03$ | $\mathbf{0.39}\pm0.02$ |
| | GPT4.1$_{nano}$ | LLM-first | $31.83\pm2.61$ | $(203.50, 203.50)\pm(23.06, 23.06)$ | $0.16\pm0.01$ | $0.14\pm0.03$ | $0.15\pm0.02$ |
| | | ILS-CSL | $31.60\pm2.06$ | $\mathbf{(147.40, 147.40)}\pm(11.25, 11.25)$ | $0.29\pm0.05$ | $0.19\pm0.04$ | $0.23\pm0.03$ |
| | | BFS | $34.75\pm7.80$ | $(252.50, 252.50)\pm(23.36, 23.36)$ | $0.18\pm0.08$ | $0.12\pm0.04$ | $0.14\pm0.03$ |
| | | BFS$_{corr}$ | $32.30\pm4.16$ | $(301.10, 301.10)\pm(14.42, 14.42)$ | $0.21\pm0.03$ | $0.15\pm0.01$ | $0.18\pm0.01$ |
| | | BLANCE | $\mathbf{27.90}\pm1.61$ | $\mathbf{(143.50, 165.50)}\pm(12.23, 35.97)$ | $\mathbf{0.42}\pm0.06$ | $\mathbf{0.23}\pm0.04$ | $\mathbf{0.30}\pm0.03$ |
| USER LEVEL DATA-I | Llama3.1$_{8B\text{-}instruct}$ | LLM-first | $11.50\pm0.50$ | $(42.50, 42.50)\pm(1.50, 1.50)$ | $0.74\pm0.01$ | $0.61\pm0.03$ | $0.67\pm0.02$ |
| | | ILS-CSL | $9.75\pm1.64$ | $(32.25, 32.25)\pm(3.56, 3.56)$ | $0.78\pm0.05$ | $0.66\pm0.03$ | $0.72\pm0.04$ |
| | | BLANCE | $\mathbf{7.50}\pm0.45$ | $\mathbf{(25.80, 25.40)}\pm(2.32, 4.22)$ | $\mathbf{0.92}\pm0.07$ | $\mathbf{0.69}\pm0.01$ | $\mathbf{0.79}\pm0.01$ |
| | Qwen3$_{4B\text{-}instruct}$ | LLM-first | $23.67\pm1.11$ | $(64.33, 64.33)\pm(0.94, 0.94)$ | $0.35\pm0.04$ | $0.28\pm0.04$ | $0.31\pm0.04$ |
| | | ILS-CSL | $20.60\pm4.18$ | $(56.80, 56.80)\pm(7.36, 7.36)$ | $0.45\pm0.13$ | $0.36\pm0.10$ | $0.40\pm0.12$ |
| | | BLANCE | $\mathbf{9.80}\pm2.04$ | $\mathbf{(26.60, 26.60)}\pm(6.83, 6.83)$ | $\mathbf{0.91}\pm0.08$ | $\mathbf{0.54}\pm0.08$ | $\mathbf{0.67}\pm0.07$ |
| | GPT4.1$_{nano}$ | LLM-first | $19.42\pm1.14$ | $(72.10, 72.10)\pm(5.36, 5.36)$ | $0.76\pm0.07$ | $0.32\pm0.03$ | $0.45\pm0.03$ |
| | | ILS-CSL | $18.00\pm1.41$ | $(54.00, 54.00)\pm(2.53, 2.53)$ | $0.53\pm0.05$ | $\mathbf{0.42}\pm0.03$ | $0.47\pm0.04$ |
| | | BFS | $36.60\pm8.96$ | $(65.80, 65.80)\pm(2.17, 2.17)$ | $0.13\pm0.06$ | $0.29\pm0.14$ | $0.18\pm0.06$ |
| | | BFS$_{corr}$ | $24.00\pm6.24$ | $(42.20, 42.20)\pm(6.53, 6.53)$ | $0.17\pm0.12$ | $0.28\pm0.11$ | $0.21\pm0.11$ |
| | | BLANCE | $\mathbf{10.60}\pm2.13$ | $\mathbf{(34.80, 35.20)}\pm(5.56, 5.84)$ | $\mathbf{0.97}\pm0.04$ | $0.43\pm0.06$ | $0.60\pm0.06$ |

show that for both CHILD and USER LEVEL DATA - I, BLANCE beats BFS handily. To elucidate about memorization, we compare the difference-of-differences in metric-values between BFS and BLANCE. The difference is *smaller* for CHILD dataset, a standard causal discovery benchmark that is likely to be a part of the LM training data. In contrast, the difference is *much larger* for the USER LEVEL DATA-I dataset. The latter dataset or its causal graph is unlikely to have been seen by any LM due to the processing and construction of attributes we performed in this dataset, from the publicly available large data (Google & Kaggle, 2018). That BFS performs worse highlights its reliance on memorized knowledge, whereas BLANCE demonstrates greater robustness across datasets.

## 6 DISCUSSION AND CONCLUSION

We present BLANCE (LM-Augmented Causal Estimation)—a Bayesian framework for causal structure discovery and parameter (edge weights) estimation in sequential, batch-wise data settings. By treating language models (LMs) as noisy surrogate experts, BLANCE addresses the dual *LM-induced* and *data-induced* biases. A key contribution is the representation shift from DAGs to PAGs, allowing uncertainty and confounders to be modeled explicitly. Through LM interactions modeled in sequential optimization framework and EM style iterative parameter estimation algorithm, BLANCE improves both structural accuracy and parameter recovery in hybrid *data-LM* pipelines. BLANCE leverages global LM knowledge while staying grounded in local data, and offers a robust foundation for hybrid causal discovery in the presence of sequential, batched data.

**Limitations and Future Work:** For the parameter estimation part, but not for causal structure discovery, BLANCE is grounded in linear SEM, which simplifies the E-step but limits generality. Extending to non-linear SEMs requires approximate inference methods, such as variational inference, which we leave for future work. In the Bayesian formulation, incorporating Dirichlet or hierarchical Bayesian priors over LM judgments could provide a principled way to capture LM uncertainty. Future work may also explore adaptive calibration of LM responses or memory-based accumulation of observational data across batches (Chang et al., 2023). We employ FCI for causal discovery due to its compatibility with our setup (Section 2). Systematically evaluating the robustness of BLANCE for other causal discovery algorithms remains important future work. Additionally, extending our approach to incorporate interventional data and active learning strategies could improve sample efficiency and the quality of discovered causal structures. Systematic evaluation of LM accuracy in confounder identification across diverse domains also remains a useful future effort. Finally, a theoretical analysis of sequential optimization and a bandit-style framework remains an open problem, discussed preliminarily in App. E.

## REPRODUCIBILITY STATEMENT

We have taken several steps to ensure the reproducibility of our results. All language model (LM) prompts used in the experiments are provided in App. G.2. The datasets employed are publicly available, with details of data processing and the simulation setup for sequential batch data described in App. B. The details about the evaluation metrics are outlined in App. D, and complete experimental details, including hyperparameters, are presented in App. G. Together, these resources enable full reproducibility of our experiments.

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

## APPENDICES

We organize the appendix as follows: App. A discusses related literature and how BLANCE differs from prior art; App. B provides detailed descriptions of all datasets used and simulation details; App. C discusses the choice of FCI over other causal discovery algorithms; App. D outlines the evaluation metrics employed to assess method performance; App. E discusses the connection between the proposed sequential optimization approach and the multi-armed bandit framework; App. F describes the parameter estimation algorithm and its robustness to LM-derived information; and App. G presents the experimental setup along with additional implementation and hyperparameter details.

## THE USE OF LARGE LANGUAGE MODELS (LLMs)

In this paper, LLMs were used only for minor grammatical edits, word polishing, and rephrasing. LLMs did not contribute in any way to research ideation, experiments, or core writing. All suggestions from LLMs were manually verified and edited by the authors prior to final inclusion.

## A LITERATURE REVIEW

**Traditional causal discovery** Traditional causal discovery aims to recover the underlying causal structure from observational data by exploiting statistical dependencies, often formalized through graphical models such as DAGs and PAGs (Spirtes & Zhang, 2016; Pearl, 2009; Neapolitan et al., 2004). These approaches include constraint-based methods (*e.g.,* PC (Spirtes et al., 2000), FCI (Spirtes et al., 1995; Zhang, 2008), RFCI (Colombo et al., 2012)), score-based methods (*e.g.,* GES (Chickering, 2002)), and functional causal models (*e.g.,* LiNGAM (Shimizu et al., 2006; 2011)), typically assuming causal sufficiency and faithfulness (Spirtes et al., 2000; Zhang & Spirtes, 2012). They rely on conditional independence tests or likelihood-based scoring to causal relationships (Shah & Peters, 2020; Zhang et al., 2011; Peters et al., 2017; Glymour et al., 2019). However, these approaches often struggle under data scarcity, presence of latent confounders (Spirtes et al., 2000; Monti et al., 2020), or domain-specific constraints not captured by statistical patterns (Mooij et al., 2016; Peters et al., 2014). These limitations are addressed by hybrid methods incorporating external domain knowledge (Meek, 1995; Heckerman et al., 1995; Ogarrio et al., 2016) from human experts, refining causal graphs with new variables, modifying edge orientations, or resolving equivalence classes (Brouillard et al., 2020; Wang et al., 2017; Constantinou et al., 2023; Ban et al., 2023b). This helps restrict the search space and improving identifiability, particularly when data is limited (Wallace et al., 1996). The evolution from purely statistical methods to knowledge-augmented approaches fuels advanced machine learning techniques to enhance causal discovery (Glymour et al., 2019; Schölkopf et al., 2021). Also, causal research distinguishes general *vs.* local settings (Baldi & Shahbaba, 2020) and applies to diverse fields (Andrade & Zachariadis, 2016; Bilal & Känzig, 2024; Geist & Lambin, 2002; Kelly et al., 2011; McKinney et al., 2016; Mathers et al., 2009).

Causal discovery extends to streaming data of networks, stock markets, and sensor systems. Causal Bayesian learning (Darvish Rouhani et al., 2018), causal discovery with progressively streaming features (Yu et al., 2010; You et al., 2023; Li et al., 2021; You et al., 2021; Yu et al., 2010) are well studied. In contrast, we focus on *contemporaneous* relationships among a fixed set of variables, which yield data across batches. Batch-wise experimentation in medicine and A/B testing examine associations between batches, batch effects as causal effects (Bridgeford et al., 2021) and adaptive batch-wise intervention (Zhang & Yuan, 2024). We confine to sequential, batch-wise observational data.

**LM-augmented causal discovery** LM augmented causal discovery relies on its world knowledge and includes pairwise prompting (Willig et al., 2022; Long et al., 2022; Kiciman et al., 2024; Jin et al., 2024; Long et al., 2023), triplet-based prompting incorporating voting (Vashishtha et al., 2025). Hybrid frameworks integrate LM-generated insights in constraint-based methods or inform score-based approaches (Ban et al., 2023b; Takayama et al., 2025). Reliability of LM-derived constraints is sensitive to domain specificity and prompt framing (Kiciman et al., 2024; Ji et al., 2023). Iterative refinement of causal graphs by alternating between LM reasoning and statistical verification is done in ILS-CSL (Ban et al., 2023a). Augmenting LM prompts with correlation matrices or statistical summaries is explored (Jiralerspong et al., 2024a; Susanti & Färber, 2025). Yet, LM-aided causal discovery shows inconsistent judgments across prompting strategies, difficulties with grounding, and

biases. Another recently proposed method is BFS (da Silva et al., 2025). However, BFS is designed for full dataset while BLANCE is anchored in sequential batched data. Additionally, BFS updates is around information gain, and it does not perform edge-weight estimation.

**Parameter estimation in SEMs** In Structural Equation Models, under assumptions of linearity and Gaussian noise (Bollen, 1989; Shimizu et al., 2006), classical approaches estimate parameters using maximum likelihood or two-stage least squares methods (Pearl, 2009). Learning of SEM parameters extends to nonlinear or nonparametric settings using neural networks or Gaussian processes, enabling flexible modeling of complex dependencies while retaining causal interpretability (Zheng et al., 2020; Lachapelle et al., 2020), and helps in integrating expert knowledge with data-driven estimation(Peters et al., 2017; Schölkopf et al., 2021). However, use of LM-derived priors and correlations for SEM parameter estimation remains largely unexplored.

Deviating from the prior art, we present a Bayesian framework for causal discovery, where data arrive sequentially in batches, and we recognize dual uncertainties—arising from limited observational data and noisy LM responses. We depart from DAG-centric discovery to adopt the PAG, a more robust representation for evolving, uncertain causal structures. We iteratively refine the PAG across batches, while framing LM queries as a sequential optimization problem. Finally, we address an under-explored aspect of hybrid causal discovery by developing a method for SEM parameter estimation that integrates LM-derived noisy priors, enabling consistent inference of causal strengths.

## B    DATASET AND SIMULATION DETAILS

EARTHQUAKE   The EARTHQUAKE dataset is a widely used synthetic benchmark in causal discovery. It comprises a small, acyclic, and semantically meaningful graph over five binary variables: *Burglary*, *Earthquake*, *Alarm*, *JohnCalls*, and *MaryCalls*. In this structure, an earthquake or burglary can trigger the alarm, which in turn influences whether John or Mary calls. The ground-truth causal graph is depicted in Fig. A1a. For our experiments, we aggregate data from six CSV files (`earthquake_250_1-6.csv`) available at `https://github.com/andrewli77/MINOBS-anc/`, resulting in a combined dataset of 1500 samples. As the dataset contains Boolean values (Yes/No) we convert them to a numerical format (1/0) producing the full dataset $\mathbb{R}^{(1500,5)}$.

ASIA   The ASIA dataset is another canonical benchmark in causal discovery and probabilistic inference. It consists of eight binary variables (*e.g., VisitToAsia*, *Tuberculosis*, *Smoking*, *LungCancer*) structured in a directed acyclic graph (DAG) with known semantics, as shown in Fig. A1b. We construct the dataset by merging all samples from the six CSV files (`asia_250_1-6.csv`) provided at `https://github.com/andrewli77/MINOBS-anc/`, yielding a total of 1500 samples. As with the EARTHQUAKE dataset, all boolean values are converted to numerical format producing the full dataset $\mathbb{R}^{(1500,8)}$.

USER LEVEL DATA - I   The USER LEVEL DATA - I dataset is derived from the publicly available Google Analytics Sample Dataset, accessible via Google BigQuery (Google & Kaggle, 2018). It captures real-world e-commerce interactions from the Google Merchandise Store. We focus on a curated subset of user-level features that reflect engagement, browsing behavior, and purchase intent. The selected variables include *Proximity to Transaction*, *Number of Add To Cart*, *Number of Product Clicks*, *Number of Sessions on iOS*, *Number of Promo Hits (Android)*, *Number of Cheap Products Viewed*, *Number of Page Hits*, *Time Spent per Session*, and *Number of Promo Hits (Others)*. Since the true causal structure is not available, we estimate it using the DirectLiNGAM algorithm applied to the observational data, which also provides edge weights representing causal strengths. We then simulate data from this estimated DAG using a structural equation model (SEM) with linear relationships and additive Gaussian noise (abiding by Assumption 5). Fig. A2 shows the causal structure with strengths estimated by DirectLiNGAM which acts as a ground truth for us.

USER LEVEL DATA - II   The USER-LEVEL DATA II dataset is constructed from the same Google Analytics Sample corpus as USER-LEVEL DATA I, but emphasizes a distinct set of behavioral features associated with user engagement. The selected variables include: *Number of Hits*, *Number of Unique URLs*, *Time Spent*, *Total Active Days Last Month*, *Number of Sessions Last Month*, *Number of Hits Last Month*, *Number of Page Hits Last Month*, and *Number of Hits on Social Network*. These features capture both cumulative and recent patterns of user interaction, providing a rich view of user behavior. Since the true causal structure is not available, as with USER-LEVEL DATA I, we

Table A1: Glossary of notations used throughout the paper, categorized by their role in datasets, causal structures, parameters, expert interactions, and other components.

| Symbol | Description | Symbol | Description | Symbol | Description |
|---|---|---|---|---|---|
| **Dataset and Observations** | | | | | |
| $\mathbb{D}$ | True data distribution | $\mathcal{D}$ | Full observed dataset | $\mathbf{D}_i$ | Observed data at batch $i$ |
| $n_i$ | Number of data points in batch $i$ | $\mathbf{V}^{\mathrm{O}}$ | Set of observed variables | $d$ | Cardinality of $\mathbf{V}^{\mathrm{O}}$ |
| $\mathbf{V}^{\mathrm{L}}$ | Unobserved (latent) variables | $k$ | Look-back batch window size | $\mathcal{B}_i$ | Background knowledge till batch $i$ |
| **Causal Graphs and Structures** | | | | | |
| $\mathcal{G}$ | True underlying causal graph | $\mathbf{V}, \mathbf{E}$ | Nodes and edges in $\mathcal{G}$ | $\mathcal{G}^{\mathrm{D}_i}$ | Causal graph after batch $i$ |
| $\mathbf{V}^{\mathrm{O}}, \mathbf{E}^{\mathrm{D}_i}$ | Nodes/edges in $\mathcal{G}^{\mathrm{D}_i}$ | $\mathcal{G}^{\mathrm{X}_i}$ | Expert-provided graph till batch i | $\mathbf{V}^{\mathrm{X}_i}, \mathbf{E}^{\mathrm{X}_i}$ | Nodes and edges in $\mathcal{G}^{\mathrm{X}_i}$ |
| $A, B$ | Arbitrary variables in graph | $L$ | Latent confounder in causal structure | | |
| **Parameters and Models** | | | | | |
| $\boldsymbol{\theta}$ | Edge weights of the causal DAG | $\boldsymbol{\theta}^{\mathrm{O}}, \boldsymbol{\theta}^{\mathrm{L}}$ | Edge weights for observed, latent variable | $\phi$ | SEM parameters $\{\boldsymbol{\theta}, \sigma^2\}$ |
| $\theta_A$ | Edge weights associated with $A$ | $\phi_A$ | Parameters for $A$ variable $(\theta_A, \sigma^2)$ | $\epsilon$ | SEM noise |
| $\sigma^2$ | SEM noise variance | $\rho(A, B)$ | Correlation value between $A, B$ | | |
| **Expert Interaction and Histograms** | | | | | |
| $\alpha$ | Expert noise level | $\mathcal{H}^{\mathbf{E}_i}$ | Edge histogram till batch $i$ | $\mathcal{H}^{\mathrm{L}_i}$ | Latent variable histogram till batch $i$ |
| $\mathcal{I}^{\mathbf{E}}, \mathcal{I}^{\mathrm{L}}$ | Instructions for edge/latent expert | $\mathcal{I}^{\mathrm{P}}, \mathcal{I}^{\rho}$ | Instructions for prior/correlation expert. | $m^{\mathbf{E}}, m^{\mathrm{L}}$ | Maximum expert calls for edge/latent |
| **Others** | | | | | |
| $\mathbb{R}$ | Set of real numbers | $\mathcal{N}$ | Gaussian distribution | $\eta$ | Learning rate |
| $\lambda$ | Regularizer parameter | $\boldsymbol{m}_p$ | Prior mean | $\boldsymbol{S}_p$ | Prior covariance |

estimate the underlying causal structure using DirectLiNGAM and simulate data from the resulting DAG using a linear SEM with additive Gaussian noise. This serves as the true causal structure for subsequent evaluation and experimentation. Fig. A3 shows causal structure with strengths estimated by DirectLiNGAM which acts as a ground truth for us.

CHILD The CHILD dataset is a widely used synthetic benchmark in causal discovery. It comprises a acyclic, and semantically meaningful graph over 20 variables: *Age, Birth Asphyxia, Disease Type, Sickness, Hypoxia in $O_2$, Grunting, Lung Parenchyma, Lower Body $O_2$, $CO_2$, Chest X-Ray, LVH, Cardiac Mixing, Birth Defect, Pulmonary Stenosis, Duct flow, Cyanosis, Heart Disorder, Temperature, Heart-Rate*, and *PV SAT*. The ground-truth causal graph is shown in Spiegelhalter et al. (1993, Figure 2). For our experiments, we aggregate data from six CSV files (child_2000_1-6.csv) available at https://github.com/andrewli77/MINOBS-anc/, resulting in a combined dataset of 12000 samples. We convert all the variables to a categorical format producing the full dataset $\mathbb{R}^{(12000, 20)}$.

ALARM The ALARM dataset is a widely used synthetic benchmark in causal discovery. It comprises a acyclic, and semantically meaningful graph over 37 variables. For our experiments, we aggregate data from the CSV files (alarm_1000_1-6.csv) available at https://github.com/andrewli77/MINOBS-anc/. All the variables are converted to categorical format and the details about it can be found in Beinlich et al. (1989).

RED WINE QUALITY The RED WINE QUALITY dataset is a real-world dataset from the UCI Machine Learning Repository, comprising 11 physicochemical attributes of red wine samples—such as *density*, *alcohol*, and *residual sugar*—along with a quality rating. We utilize this dataset for *parameter estimation* experiments. As shown in Fig. A5, the variable *alcohol* acts as a confounder between *density* and *quality*, making it suitable for parameter estimation evaluation. The dataset used in our experiments is sourced from the CMU Example Causal Datasets repository: https://github.com/cmu-phil/example-causal-datasets/blob/main/real/wine-quality/data/winequality-red.continuous.txt. It contains a total of 1599 samples.

## C COMPARISONS WITH OTHER CAUSAL DISCOVERY ALGORITHMS

In this paper, we focus on a setting where both *selection bias* and *latent confounding* are present (*cf.* Assumption 3). To accommodate both aspects in this general setting, it is necessary to shift from DAGs to Partial Ancestral Graphs (PAGs), since this explicitly captures uncertainty due to latent confounders and selection bias. The Fast Causal Inference (FCI) algorithm naturally aligns with these assumptions, making it a principled choice for the proposed framework, BLANCE.

Moreover, the primary aim of this paper is not to benchmark causal structure discovery algorithms, but rather to demonstrate how BLANCE can refine causal structure over time by integrating noisy expert

knowledge (from LMs) with observational data. Notably, BLANCE is designed to be algorithm-agnostic, and can be used in a plug-and-play manner with any PAG-generating causal discovery method. Below, we briefly discuss why several modern algorithms are not directly applicable:

1. **Greedy Equivalence Search (GES):** GES assumes both causal sufficiency and the absence of selection bias. These assumptions do not hold in our sequential setting, where both latent confounders and selection bias may be present.

2. **DAG-NoCURL and DAG-NoTEARS:** These are continuous optimization-based methods that output DAGs under the assumption of causal sufficiency, without accounting for latent confounders. Thus, they do not apply to our setting, which requires a PAG-based representation to model uncertainty.

3. **DAGMA:** DAGMA is a score-based causal discovery method that also formulates the problem as a continuous optimization task. Like DAG-NoCURL and DAG-NoTEARS, it outputs deterministic DAGs and assumes no latent confounding, making it incompatible with the sequential setting.

4. **LiNGAM:** LiNGAM (Linear Non-Gaussian Acyclic Model) assumes that all relevant variables are observed, *i.e.* there are no hidden confounders. This causal sufficiency assumption renders it unsuitable for the sequential setting.

5. **Recursive Causal Discovery (RCD):** Unlike FCI, RCD produces a DAG rather than a PAG, focusing on efficient DAG learning. Hence, when dealing with latent confounders and selection bias requiring PAG representation, RCD can not be employed.

We note that enhanced variants of FCI such as RFCI, GFCI, and FCI+ are fully compatible with BLANCE and can be seamlessly integrated bringing their advantage to BLANCE as well. We keep the research to study convergence impact of different algorithms for future work.

## D  EVALUATION METRICS

We describe the evaluation metrics used to assess the performance of the causal discovery method.

### D.1  MODIFIED STRUCTURAL HAMMING DISTANCE (MOD. SHD)

To evaluate the structural accuracy of methods, we use a modified version of the Structural Hamming Distance (SHD) tailored for Partial Ancestral Graphs (PAGs). Unlike standard causal graphs, PAGs include multiple edge types that reflect varying degrees of causal certainty. This necessitates a more nuanced treatment of structural differences.

For example, consider the true edge $A \rightarrow B$. If one predicted PAG contains $A \leftrightarrow B$ (a definite bidirected edge) and another contains $A \circ\!\!\rightarrow B$ (an edge with uncertainty at one endpoint), the former should incur a higher penalty since it reflects stronger, incorrect causal commitment. That is,

$$\text{Mod. SHD}(A \rightarrow B, A \leftrightarrow B) > \text{Mod. SHD}(A \rightarrow B, A \circ\!\!\rightarrow B). \tag{A1}$$

Standard SHD counts the total number of missing edges, extra edges, and orientation mismatches. Our modified SHD refines the orientation mismatch term by assessing the endpoints of each edge separately. We assign a penalty of $1.0$ for definite orientation errors (*e.g.*, $\rightarrow$ vs. $\leftarrow$) and $0.5$ for uncertain mismatches (*e.g.*, $\rightarrow$ vs. $\circ\!\!\rightarrow$). This weighting scheme better reflects the confidence associated with different edge types and penalizes definitive errors more heavily than uncertainty.

### D.2  STRUCTURAL INTERVENTION DISTANCE (SID)

The Structural Intervention Distance (SID) measures the robustness of a learned causal graph in supporting correct interventional reasoning. Unlike purely structural metrics such as SHD, SID evaluates whether the predicted graph implies the correct set of causal effects under interventions.

Formally, SID counts the number of intervention targets for which the predicted graph induces an incorrect adjustment set relative to the true graph. An SID of zero indicates that the learned

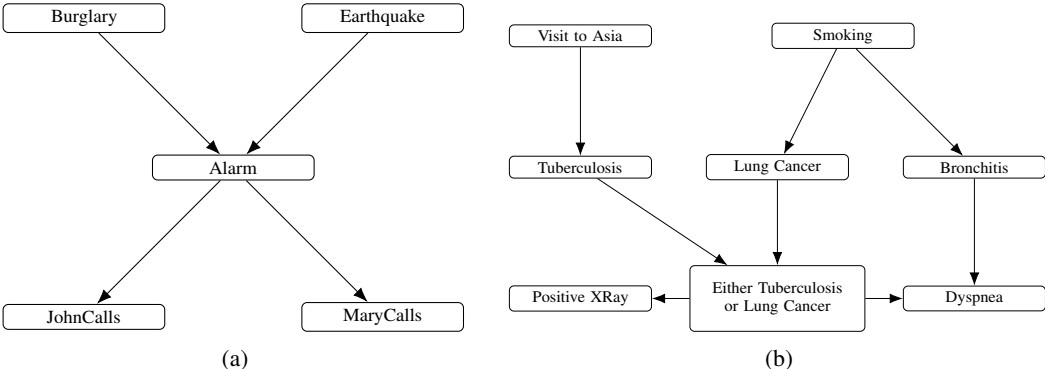

Figure A1: **Causal Structure Learning Datasets** *(a)* Ground-truth causal graph for the EARTH-QUAKE dataset, where an earthquake or burglary can trigger an alarm, which in turn causes calls from John and Mary; *(b)* Ground-truth structure for the ASIA dataset, a classic dataset illustrating causal relations between variables such as smoking, lung disease, and visits to Asia.

structure supports identically correct interventional distributions as the ground truth, even if some edge directions are incorrect but do not affect adjustment validity.

SID is particularly useful in settings where downstream causal queries, rather than exact graph recovery, are the primary concern. It captures whether the learned structure preserves the correct set of (in)dependencies required for estimating interventional effects, making it a practical and task-aligned evaluation metric for causal discovery.

### D.3 PRECISION, RECALL, F1-SCORE

In addition to structural metrics, we report standard classification metrics—Precision, Recall, and F1-Score—computed over predicted directed causal relationships of the form $A \rightarrow B$.

A predicted edge $A \rightarrow B$ is considered a true positive if it matches a directed edge in the ground-truth causal graph. Precision measures the fraction of predicted directed edges that are correct, while Recall quantifies the fraction of ground-truth directed edges that are successfully recovered. F1-Score is the harmonic mean of Precision and Recall, providing a balanced summary of accuracy and completeness:

$$\text{Precision} = \frac{|\text{Correctly predicted directed edges } A \rightarrow B|}{|\text{Predicted directed edges } A \rightarrow B|}, \tag{A2}$$

$$\text{Recall} = \frac{|\text{Correctly predicted directed edges } A \rightarrow B|}{|\text{Ground-truth directed edges } A \rightarrow B|}, \tag{A3}$$

$$\text{F1 Score} = 2 \cdot \frac{\text{Precision} \cdot \text{Recall}}{\text{Precision} + \text{Recall}}. \tag{A4}$$

These metrics focus exclusively on directed edges and ignore uncertain or undirected edge types. As such, they offer a more task-specific evaluation of causal directionality recovery, which is critical in downstream applications that rely on explicit causal mechanisms.

### E SEQUENTIAL OPTIMIZATION AS MULTI-ARMED BANDIT

The LM interaction in BLANCE proposed as sequential optimization (*cf.* Sec. 3.3), can presumably be interpreted as a *stochastic multi-armed bandit (MAB)* problem, providing a useful lens for understanding the design of the proposed score selection strategy (Eq. (9)).

**From Edge Querying to Bandits**  In MAB problems, a learner chooses among $K$-*arms*, each associated with an unknown reward distribution, and aims to maximize cumulative reward over $T$ pulls by balancing *exploration* (learning about uncertain arms) and *exploitation* (favoring known high-reward arms).

In our setting (as discussed in Eq. (6)):

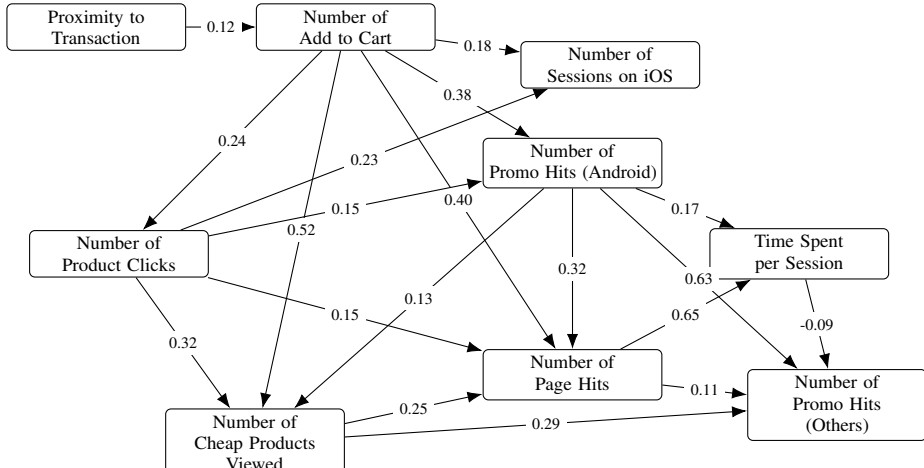

Figure A2: Ground truth causal structure for the USER LEVEL DATA - I illustrating the causal relationships between user behavioral metrics. The edge weights represent estimated causal strengths derived from observational data.

- Each candidate edge $e$ is treated as an **arm**.

- Pulling arm $e$ means **querying the LM** about edge $e$'s causal type.

- **Reward** is the **information gain** from that query—specifically, the reduction in uncertainty over edge $e$'s type.

- Total LM query budget $m_L$ acts as the $T$ **horizon**.

This framing is motivated by the need to *adaptively allocate* a limited number of LM queries to edges that are either currently uncertain (exploration) or close to being included in background knowledge (exploitation). As in bandits, where arms may yield noisy feedback, LM responses are noisy, that is, stochastic and depend on context and prompt. Hence, each edge must be queried multiple times to obtain a reliable posterior.

### E.1 UCB-INSPIRED SELECTION SCORE

The selection score proposed in Eq. (9) resembles a classic *Upper Confidence Bound (UCB)* strategy,

$$S_i^e = w_1 E_i^e + w_2 \left( \frac{1}{TD_i^e} \right) + w_3 \sqrt{\frac{\log T_i}{T_i^e}}, \quad \text{s.t.} \quad TD_i^e = \tau_i - \max(\mathcal{H}^{\mathbf{E}_i}(e)). \quad \text{(A5)}$$

This selection score balances three factors:

- **Uncertainty (Exploration):** $E_i^e$ is the entropy of the edge's histogram—capturing how uncertain the current belief over $e$ is. High entropy indicates high uncertainty, encouraging exploration.

- **Threshold Proximity (Exploitation):** $1/TDe_i$ quantifies how close edge $e$ is to being included in background knowledge. A small $TDe_i$ yields a large score, encouraging exploitation of promising edges.

- **Exploration Bonus:** The term $\sqrt{\log T_i/T_i^e}$ mirrors the optimism term in UCB1. It grows slowly with total queries and shrinks with more queries to edge $e$, thus favoring edges that have not been sampled much.

Together, this scoring function implements a principled query policy that prioritizes under-explored, informative, and nearly-threshold edges. Each LM query corresponds to a pull of the edge with the highest $S_i^e$, analogous to greedy selection in UCB-based bandits.

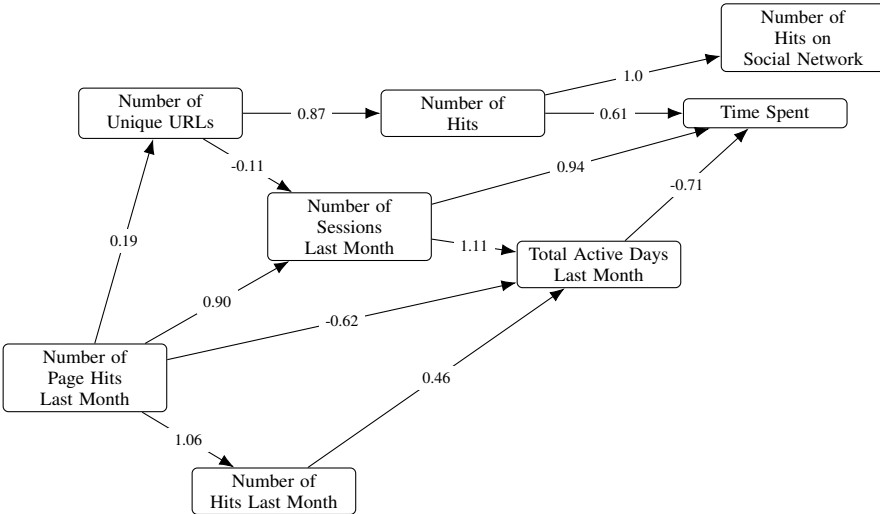

Figure A3: Ground truth causal structure for the USER LEVEL DATA - II illustrating the causal relationships between user activity metrics such as hits, time spent, session history, and social network interactions. The edge weights represent estimated causal strengths derived from observational data.

### E.2 REGRET INTERPRETATION

Under idealized assumptions of *i.i.d.* edge-level information gains, bounded support, and stationary LM behavior, the expected cumulative *regret* after $T$ queries is:

$$\mathcal{R}(T) = \sum_{i=1}^{T} (\mu^* - \mu_{a_i}), \tag{A6}$$

where $\mu^*$ is the maximum expected gain over all edges, and $a_i$ is the edge queried at step $i$. Classical UCB algorithms ensures that

$$\mathcal{R}(T) = O\left(\sum_{e:\Delta_e>0} \frac{\log T}{\Delta_e}\right), \quad \text{where } \Delta_e = \mu^* - \mu_e. \tag{A7}$$

This sublinear growth implies that the average regret $\mathcal{R}(T)/T \to 0$ as $T \to \infty$, *i.e.* the learner asymptotically focuses on optimal arms.

Applied to BLANCE, this suggests that if LM responses were non-stochastic, the selection score strategy concentrates LM calls on most informative edges, making increasingly efficient use of limited expert budget. However, we **caution** against this linkage to regret bounds. The above analysis rests on assumptions that do not strictly hold in the proposed BLANCE setting due to:

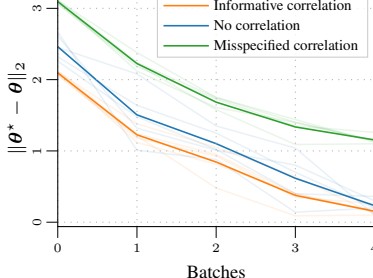

- **Stochasticity:** The LM's behavior is batch-context dependent and evolves as context accumulates. Reward distributions (information gains) are not fixed.

- **Dependent arms:** Updates to the posterior of one edge can influence others due to graph constraints, violating arm independence.

- **Implicit priors:** LM outputs are influenced by prompts, prior batches, and temperature, introducing structured, non-*i.i.d.* noise.

Figure A4: **Parameter Estimation:** Convergence of parameters with LM-suggested correlation as more batches are processed.

These violations of MAB assumptions mean that classical regret bounds cannot be directly applied. Still, the MAB abstraction provides valuable intuition for designing and analyzing selection policies under uncertainty.

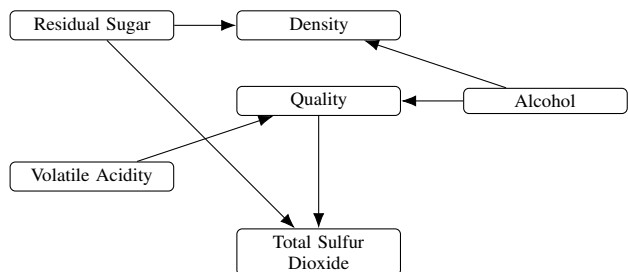

Figure A5: Ground truth causal structure for the RED WINE QUALITY dataset illustrating the directional dependencies among key physicochemical attributes affecting wine quality. The graph highlights how alcohol, volatile acidity, total sulfur dioxide, and density (influenced by residual sugar) contribute directly or indirectly to the perceived quality of red wine.

### E.3 FUTURE WORK

A full theoretical analysis of regret in this setting would require modeling *non-stationary, dependent* reward structures. Promising directions include *contextual and Bayesian bandits* that incorporate evolving priors, *combinatorial bandits* for structured edge dependencies, and *information-theoretic regret bounds* based on entropy reduction. We leave these extensions for future work.

## F PARAMETER ESTIMATION

As outlined in Sec. 4, we propose a Bayesian parameter estimation framework designed to estimate structural parameters in the presence of latent confounders. This approach incorporates language model (LM)-suggested prior knowledge into an Expectation-Maximization (EM) algorithm. The EM-step are defined in Eq. (10) and Eq. (11).

In the experiment on the RED WINE QUALITY dataset (*cf.* Sec. 5.2), we evaluate the impact of incorporating LM-suggested priors over the latent confounder. Fig. 6 illustrates how different priors affects estimation performance, demonstrating the robustness of the proposed Bayesian method.

Additionally, we analyze the effect of LM-suggested correlation values $\rho(A, B)$ and investigate the robustness of the proposed parameter estimation algorithm. From the observational data we know that the confounder *alcohol* has a correlation of $\rho(\text{alcohol}, \text{density}) = -0.50$ and $\rho(\text{alcohol}, \text{quality}) = 0.48$. We set the correlation regularizer as $\lambda = 5$.

For parameter estimation algorithm, we use stochastic gradient descent with 0.001 learning rate and run the algorithm for maximum for maximum of 20 E-steps and 50 M-steps with an early stop criteria based on the expected log-likelihood value.

## G EXPERIMENT DETAILS

This section provides detailed descriptions of the experimental setup, including sequential data setup and batching strategies, LM prompts used in the experiments, details about the FCI variants, LLM-first, ILS-CSL, and BLANCE framework.

### G.1 SEQUENTIAL DATA SETUP

We describe the setup for each dataset category based on the availability of ground-truth causal structure and the strategy used for batch construction.

EARTHQUAKE, ASIA These datasets provide both the true causal structure and observational data. We use the ground-truth graphs shown in Fig. A1a and Fig. A1b as the true causal structures. The observational data are randomly partitioned into 6 batches of 250 samples each, without replacement. This results in data of shape $(6, 250, 5)$ for EARTHQUAKE and $(6, 250, 8)$ for ASIA. Since all variables are binary (Yes/No), we preprocess the data by mapping Yes to 1 and No to 0.

USER-LEVEL DATA I, USER-LEVEL DATA II For these datasets, ground-truth causal structures are unavailable. We apply DirectLiNGAM to the full observational dataset to estimate a causal graph, which we then treat as the true causal structure for generating synthetic data. The resulting graphs are visualized in Fig. A2 and Fig. A3. Using the estimated structure, we simulate data via a linear SEM with additive Gaussian noise, sampled from $\mathcal{N}(0, 0.05)$. We construct 7 sequential batches with varying sizes. For USER LEVEL DATA I, the batchsize is $[3000, 1000, 2000, 4000, 3000, 2000, 5000]$ while for USER LEVEL DATA II we set the batch size as $[2000, 1000, 2000, 1000, 1000, 3000, 1000]$. To simulate realistic distributions, in USER-LEVEL DATA I, the parent node *Proximity To Transaction* is sampled from $\mathcal{N}(10, 1)$ while in USER-LEVEL DATA II, the parent node *Number Of Page Hits Last Month* is sampled from $\mathcal{N}(27, 10.5)$.

CHILD, ALARM These datasets provide both the true causal structure and observational data. We use the ground-truth graphs from Beinlich et al. (1989); Spiegelhalter et al. (1993) as the true causal structures. For CHILD dataset, the observational data are randomly partitioned into 6 batches of 2000 samples each, without replacement. This results in data of shape $(6, 2000, 20)$. For ALARM dataset, we split the observational data randomly, without replacement, into six batches with 50 data points per batch. This is done with the aim to mimic the data-scarce scenario and showcase the robustness of BLANCE.

RED WINE QUALITY This dataset provides observational data, which we use primarily to evaluate the proposed *parameter estimation* framework. We estimate the causal structure using DirectLiNGAM on the full dataset, as shown in Fig. A5. Existence of a latent confounder, *alcohol*, between *density* and *quality* makes this dataset suitable for testing latent-aware estimation. The full dataset consists of 1599 samples across 6 variables, which we partition into 5 batches of sizes $[319, 319, 319, 319, 323]$ on which we perform parameter estimation.

## G.2 LM PROMPTS

In this section, we provide the prompts used in the experiments.

The pairwise prompt is:

```
System message:

You are an expert in Causal discovery and are studying {
experiment_name}. You are using your knowledge to help build
a causal model that contains  all the assumptions about {
experiment_name}, where a causal model is a conceptual model
that describes the causal mechanisms of a system. You will do
 this by by answering questions about cause and effect and
using your domain knowledge as an expert in Causal discovery.
 We are considering the following variables: {variables}. The
 description of the variables is as follows:{
variable_description}.

User message:

From your perspective as an expert in Causal discovery, which
 of the following is  most likely true?
(A) {var1} affects/causes {var2}; {var1} has a high
likelihood of directly influencing {var2};
(B) {var2} affects/causes {var1};  {var2} has a high
likelihood of directly influencing {var1};
(C) Neither A nor B; There is no causal relationship between
{var1} and {var2}.
(D) Do not know about the causal relationship between {var1}
and {var2}.
Select the answer. Think step by step and provide your
thoughts with the "thoughts" key and the answer with the "
answer" key.
Return a JSON with the following format:
{
        "answer": "A/B/C/D",
```

```
                    "thoughts": "step-by-step thought"
            }
        NOTE: Only return the JSON and nothing else.
```

The triplet prompt is:

```
        System message:

        You are an expert in Causal discovery and are studying {
        experiment_name}. You are using your knowledge to help build
        a causal model that contains  all the assumptions about {
        experiment_name}, where a causal model is a conceptual model
        that describes the causal mechanisms of a system. We are
        considering the following variables: {variables}. The
        description of the variables is as follows:{
        variable_description}.

        User message:

        As an expert in Causal discovery, consider the following
        variables and output a causal DAG, {var1}, {var2}, {var3}.
        Only consider direct causal effects. If a variable has no
        causal relationship with any other, include it as an isolated
         node.
        For example, if Z is independent, and X causes/affects Y, the
         output DAG should be: [["X", "Y"], ["Z"]]. Think step by
        step and provide your thoughts with the "thoughts" key.
        Return a JSON in the following format:
        {
                "dag": [["source_node", "target_node"], ..., ["
                isloated_node"]],
                "thoughts": "step-by-step thought"
        }
        NOTE: Only return the JSON object and nothing else.
```

The PAG-Pairwise prompt is:

```
        System message:

        You are an expert in Causal discovery and are studying {
        experiment_name}. You are using your knowledge to help build
        a causal model that contains  all the assumptions about {
        experiment_name}, where a causal model is a conceptual model
        that describes the causal mechanisms of a system. You will do
         this by by answering questions about cause and effect and
        using your domain knowledge as an expert in Causal discovery.
         We are considering the following variables: {variables}. The
         description of the variables is as follows:{
        variable_description}.

        User message:

        From your perspective as an expert in Causal discovery, which
         of the following is  most likely true?
        A: {var1} affects/causes {var2}; {var1} has a high likelihood
         of directly influencing {var2};
        B: {var2} affects/causes {var1};  {var2} has a high
        likelihood of directly influencing {var1};
        C: Neither A nor B; There is no causal relationship between {
        var1} and {var2}.
        D: Do not know about the causal relationship between {var1}
        and {var2}.
        E: There is a possible latent confounder between {var1} and {
        var2} i.e. {var1} <-> {var2}
```

```
        F: Can not be sure about the causal relationship however {
        var1} is not an ancestor of {var2} i.e. {var1} o-> {var2}
        G: Can not be sure about the causal relationship i.e. {var1}
        o-o {var2}
        Select the answer. Think step by step and provide your
        thoughts with the "thoughts" key and the answer with the "
        causal_option" key.
        Return a JSON with the following format:
        {{
                    "causal_option": "A/B/C/D/E/F/G",
                    "thoughts": step-by-step thought
        }}
        NOTE: Only return the JSON and nothing else.
```

The *LM-expert* prompt to get prior over the latent confounder

```
        System message:

        You are an expert in Causal discovery and are studying {
        experiment_name}. You are using your knowledge to help
        information about the latent confounder variable in the
        causal structure. You will do this by giving a conditional
        Gaussian distribution over the detected confounder variable
        conditioned on the two connected variables and its
        correlation with the connected variables. We are considering
        the following variables: {variables}. The description of the
        variables is as follows:{variable_description}.

        User message :

        We know that there is a confounder {confounder} between {
        variable_1} and {variable_2}. Provide an informative
        conditional Gaussian on the {confounder} conditioned on {
        variable_1} and {variable_2} using historical data and world
        knolwedge.
        Think step-by-step.
        Return a JSON with the following format.
        {{
                    "mean": "numerical mean value of prior N({
                    latent} | {variable_1}, {variable_2})",
                    "variance": "numerical variance value of
                    prior N({latent} | {variable_1}, {variable_2
                    })",
                    "correlation": "dictionary of correlation
                    numerical values",
        }}
        NOTE: Only return the JSON and nothing else.
```

The *LM-expert* prompt:

```
        System message:

        You are an expert in Causal discovery and are studying {
        experiment_name}. You are using your knowledge to help build
        a causal model that contains  all the assumptions about {
        experiment_name}, where a causal model is a conceptual model
        that describes the causal mechanisms of a system. You will do
         this by by answering questions about cause and effect and
        using your domain knowledge as an expert in Causal discovery.
         We are considering the following variables: {variables}. The
         description of the variables is as follows:{
        variable_description}.

        User message:
```

```
            You are asked to determine the causal relationship between {A
            } and {B}. Only consider direct relationships and not
            indirect ones.The options available are limited to FCI output
             i.e. PAG edges (Use 0,2,3,4,6 options carefully and think
            more about 1,2,5 as we need more of these options):
            0: There is no causal relationship between {A} and {B}.
            1: Changing the state of node {A} causally affects a change
            in another node {B} i.e. A->B
            2: There is a possible latent confounder between {A} and {B}
            3: Can not be sure about the causal relationship however {A}
            is not an ancestor of {B} i.e. {A} o-> {B}
            4. Can not be sure about the causal relationship, i.e., {A} o
            -o {B} or {B} o-o {A}
            5. Changing the state of node which says {B} causally affects
             a change in another node which says {A}, i.e. B->A
            6. Can not be sure about the causal relationship however {B}
            is not an ancestor of {A}, {B} o-> {A}

            Response format:
            {
                    "option": option_tag,
                    "thoughts": "step-by-step thought"
            }
            We know the following causal relationships: {
            known_relationship}
            Be extra thoughtful and careful about the relationships you
            are describing by considering both ({A}, {B}) and ({B}, {A})
            before answering.
            NOTE: Only return the JSON object and nothing else.
```

### G.3 FAST CAUSAL INFERENCE (FCI) VARIANTS

In the interested setting where data arrive sequentially in batches, running standard causal discovery algorithms like FCI on the entire dataset is infeasible (Assumption 4) . To address this, we adapt the Fast Causal Inference (FCI) algorithm to sequential processing by designing multiple baselines that reflect different trade-offs. Each variant operates under a restricted lookback window size, and adapts the FCI procedure accordingly.

For all FCI-based methods, we set the significance level to $\alpha = 0.3$ for the USER LEVEL DATA II dataset, and $\alpha = 0.1$ for all other datasets. We use the *chisq* test for conditional independence on the ASIA and EARTHQUAKE datasets (due to their binary variables), and the *fisherz* test for all remaining experiments involving continuous data.

**FCI-Cumulative** This variant applies the FCI algorithm to the cumulative data up to batch $i$, *i.e.* $\mathcal{G}^{D_i} = \text{FCI}(\mathbf{D}_{1:i})$. It serves as an upper bound on performance, assuming full access to all prior data.

**FCI-Vanilla** This variant applies FCI independently to each incoming batch, $\mathcal{G}^{D_i} = \text{FCI}(\mathbf{D}i)$. Here, the algorithm forgets all previous knowledge, mimicking a naive local learner. This variant tests whether single-batch inference is sufficient and highlights the limitations of ignoring *data bias*.

**FCI-Iterative** This variant introduces background knowledge by passing the output of the previous batch's FCI run as input to the next, $\mathcal{G}^{D_i} = \text{FCI}(\mathbf{D}i, \mathcal{B}_i = \mathcal{G}^{D_{(i-1)}})$. Here, the background knowledge includes previously inferred causal structure. This allows FCI to refine its structure incrementally.

**FCI-Heuristics** In this variant, an edge from previous iterations is included in the background knowledge only if a heuristic threshold is met. Specifically, an edge is included in the background knowledge if it has appeared in the FCI outputs of at least $h$ of all the past batches (we set $h=2$ in our experiments). This strategy balances between overfitting to noise (as in FCI-Iterative) and excessive forgetting (as in FCI-Vanilla).

Table A2: **Causal Discovery, Impact of LM:** We experiment with GPT-4o and GPT-5 on the EARTHQUAKE and ASIA dataset and report *Modified SHD* ↓ with temperature 1. We report mean and standard deviation over 5 showcasing the LM-agnostic nature of BLANCE and superior performance across models. The inference cost of recent models, GPT-4o and GPT-5, are quite a bit more than GPT-3.5$_{turbo}$, constraining use of them for all large set of experiments.

| Dataset | Method | Mod. SHD (GPT-3.5$_{turbo}$) | Mod. SHD (GPT-4o) | Mod. SHD (GPT-5) |
|---|---|---|---|---|
| EARTHQUAKE | LLM-First | $6.00 \pm 0.82$ | $5.80 \pm 0.75$ | $5.50 \pm 0.50$ |
| | ILS-CSL | $2.38 \pm 0.96$ | $1.50 \pm 1.12$ | $1.40 \pm 1.02$ |
| | BLANCE | $1.00 \pm 0.63$ | $0.90 \pm 0.51$ | $0.80 \pm 0.74$ |
| ASIA | LLM-First | $7.33 \pm 0.94$ | $7.90 \pm 0.40$ | $7.00 \pm 1.00$ |
| | ILS-CSL | $6.50 \pm 0.50$ | $5.00 \pm 1.00$ | $5.00 \pm 0.89$ |
| | BLANCE | $4.60 \pm 1.02$ | $3.60 \pm 1.01$ | $3.20 \pm 0.98$ |

### G.4 LLM-FIRST

In the **LLM-first** method, we reverse the standard causal discovery pipeline by first using a language model (LM) to propose an initial causal graph, which is then incrementally refined using sequential batches of observational data. To obtain the initial causal structure, we use a pairwise prompting strategy where the LM is queried on each variable pair. This results in a causal structure $\mathcal{G}^{X}$, where each edge $A \rightarrow B$ indicates a predicted causal relation. While this initial graph often captures plausible structural patterns, it suffers from overly-optimistic behavior (*cf.* Table 1) and lacks data grounding. We then sequentially refine $\mathcal{G}^{X}$ using batches of observed data. At each step $i$, we update the causal graph $\mathcal{G}^{X_{(i-1)}}$ using the FCI produced causal structure conditioned on the batch $i$. This adds, removes, or reorients edges based on statistical tests applied to the current data batch $\mathcal{D}_i$, thus improving the reliability of the structure over time.

### G.5 ITERATIVE LLM-SUPERVISED CAUSAL STRUCTURE LEARNING (ILS-CSL)

The Iterative LLM-Supervised Causal Structure Learning (ILS-CSL) (Ban et al., 2023a) algorithm integrates natural language causal knowledge into the structure learning process by iteratively refining a causal graph using response of a large language model (LLM). Originally designed as a score-based method, ILS-CSL supervises structure learning by posing pairwise causal queries to the LLM and using its responses as soft constraints during graph optimization. For consistency and a fair comparison with our method, we adapt ILS-CSL to work with the FCI framework. Specifically, we treat the causal constraints inferred from the LLM as background knowledge and inject them into the FCI algorithm. This hybrid adaptation retains the core iterative supervision strategy of ILS-CSL while operating within a constraint-based causal discovery framework.

### G.6 BLANCE

BLANCE relies on a set of hyperparameters that guide expert interaction and edge selection throughout sequential batches. The selection score used for deciding which edge to query is computed using Eq. (9) with weights $w_1$, $w_2$, and $w_3$. Additionally, for background threshold Eq. (5), $\alpha$ and a maximum expert budget $m^{\mathbf{E}}$ per batch are used to regulate expert calls.

For the ASIA, EARTHQUAKE, and USER-LEVEL DATA I datasets, for selection score we set $(w_1, w_2, w_3) = (0.1, 0.6, 0.3)$, with and background knowledge threshold with $\alpha = 0.3$ and maximum expert call budgets $m^{\mathbf{E}} = 20$, $m^{\mathbf{E}} = 50$, and $m^{\mathbf{E}} = 70$ respectively. A minimum threshold of 10 is also imposed to ensure reliability. For the USER-LEVEL DATA II dataset, we use $(w_1, w_2, w_3) = (0.3, 0.4, 0.3)$, with $\alpha = 0.5$, maximum expert call budget $m^{\mathbf{E}} = 20$, and a minimum threshold of 5. Table A3 showcases the BLANCE—structure learning algorithm across LM temperatures.

**GPT-4o and GPT-5** In Table 1, we illustrate the issue of LLM over-optimism using both GPT-3.5$_{turbo}$ and GPT-4o. We find that GPT-4o does not always outperform GPT-3.5$_{turbo}$. We further demonstrate how an LM can be viewed as a noisy expert; yet can be integrated into a Bayesian causal structure discovery framework. The inference cost of recent models, GPT-4o and GPT-5, are quite a bit more than GPT-3.5$_{turbo}$, constraining use of them for all large set of experiments. That said, BLANCE is fully model-agnostic, and stronger LMs are expected to further improve BLANCE's

Table A3: **Causal Discovery, Impact of LM Temperature:** Table 3 shows evaluation metrics only for *temperature=1*; here we show for other two *temperatures={0.0,0.5}*. The conclusions are remarkably similar and favor BLANCE. We show results for all six datasets using the two paradigms: *Only-Data* and *Data-LM*. We evaluate with the following metrics: *Modified SHD, SID, Precision, Recall, F1*. All the methods use GPT-3.5$_{turbo}$ as an LM-expert and we report mean and standard deviation over 5 runs.

| Dataset | Approach | Method | Mod. SHD ↓ | SID ↓ | Precision ↑ | Recall ↑ | F1 Score ↑ |
|---|---|---|---|---|---|---|---|
| EARTHQUAKE ($d=5$) | Only-Data | FCI-Cumulative | $2.00\pm0.00$ | $(0.00, 5.00)\pm(0.00, 0.00)$ | $1.00\pm0.00$ | $0.50\pm0.00$ | $0.67\pm0.00$ |
| | | FCI-Vanilla | $3.60\pm0.80$ | $(8.20, 9.20)\pm(3.60, 1.60)$ | $0.20\pm0.40$ | $0.05\pm0.10$ | $0.08\pm0.16$ |
| | | FCI-Iterative | $5.00\pm1.67$ | $(12.20, 12.20)\pm(4.66, 4.66)$ | $0.30\pm0.27$ | $0.20\pm0.19$ | $0.24\pm0.22$ |
| | | FCI-Heuristics | $3.60\pm0.80$ | $(8.20, 9.20)\pm(3.60, 1.60)$ | $0.20\pm0.40$ | $0.05\pm0.10$ | $0.08\pm0.16$ |
| | Data-LM$_{(t=0)}$ | LLM-first | $6.00\pm0.82$ | $(15.00, 15.00)\pm(0.82, 0.82)$ | $0.11\pm0.16$ | $0.08\pm0.12$ | $0.09\pm0.13$ |
| | | ILS-CSL | $2.00\pm0.89$ | $(5.00, 5.00)\pm(0.89, 0.89)$ | $0.93\pm0.13$ | $0.55\pm0.19$ | $0.67\pm0.17$ |
| | | BLANCE | $1.00\pm0.63$ | $(2.20, 2.20)\pm(1.60, 1.60)$ | $1.00\pm0.00$ | $0.75\pm0.16$ | $0.85\pm0.11$ |
| | Data-LM$_{(t=0.5)}$ | LLM-first | $6.00\pm0.82$ | $(15.00, 15.00)\pm(0.82, 0.82)$ | $0.11\pm0.16$ | $0.08\pm0.12$ | $0.09\pm0.13$ |
| | | ILS-CSL | $2.00\pm0.89$ | $(5.00, 5.00)\pm(2.61, 2.61)$ | $0.93\pm0.13$ | $0.55\pm0.19$ | $0.67\pm0.17$ |
| | | BLANCE | $1.00\pm0.63$ | $(2.20, 2.20)\pm(1.60, 1.60)$ | $1.00\pm0.00$ | $0.75\pm0.16$ | $0.85\pm0.11$ |
| ASIA ($d=8$) | Only-Data | FCI-Cumulative | $7.00\pm0.00$ | $(23.00, 49.00)\pm(0.00, 0.00)$ | $0.00\pm0.00$ | $0.00\pm0.00$ | $0.00\pm0.00$ |
| | | FCI-Vanilla | $7.80\pm0.75$ | $(30.00, 35.00)\pm(5.90, 2.45)$ | $0.00\pm0.00$ | $0.00\pm0.00$ | $0.00\pm0.00$ |
| | | FCI-Iterative | $8.00\pm1.26$ | $(33.00, 33.00)\pm(7.46, 7.46)$ | $0.45\pm0.24$ | $0.23\pm0.15$ | $0.29\pm0.18$ |
| | | FCI-Heuristics | $7.80\pm0.75$ | $(30.00, 35.00)\pm(5.90, 2.45)$ | $0.00\pm0.00$ | $0.00\pm0.00$ | $0.00\pm0.00$ |
| | Data-LM$_{(t=0)}$ | LLM-first | $7.33\pm0.94$ | $(27.67, 27.67)\pm(2.49, 2.49)$ | $0.58\pm0.12$ | $0.29\pm0.06$ | $0.39\pm0.08$ |
| | | ILS-CSL | $5.20\pm0.75$ | $(21.80, 21.80)\pm(1.94, 1.94)$ | $0.90\pm0.10$ | $0.38\pm0.08$ | $0.53\pm0.09$ |
| | | BLANCE | $4.60\pm1.02$ | $(13.60, 13.60)\pm(3.83, 3.83)$ | $0.80\pm0.12$ | $0.60\pm0.12$ | $0.67\pm0.08$ |
| | Data-LM$_{(t=0.5)}$ | LLM-first | $7.33\pm0.94$ | $(27.67, 27.67)\pm(2.49, 2.49)$ | $0.58\pm0.12$ | $0.29\pm0.06$ | $0.39\pm0.08$ |
| | | ILS-CSL | $5.60\pm1.36$ | $(22.80, 22.80)\pm(3.19, 3.19)$ | $0.85\pm0.15$ | $0.35\pm0.09$ | $0.49\pm0.12$ |
| | | BLANCE | $4.60\pm1.02$ | $(13.60, 13.60)\pm(3.83, 3.83)$ | $0.80\pm0.12$ | $0.60\pm0.12$ | $0.67\pm0.08$ |
| USER LEVEL DATA-I ($d=9$) | Only-Data | FCI-Cumulative | $15.00\pm2.77$ | $(47.80, 47.80)\pm(4.62, 4.62)$ | $0.72\pm0.18$ | $0.37\pm0.09$ | $0.47\pm0.09$ |
| | | FCI-Vanilla | $21.30\pm3.50$ | $(61.60, 61.60)\pm(5.54, 5.54)$ | $0.41\pm0.18$ | $0.22\pm0.10$ | $0.29\pm0.13$ |
| | | FCI-Iterative | $5.60\pm1.20$ | $(23.40, 23.40)\pm(7.94, 7.94)$ | $0.93\pm0.04$ | $0.77\pm0.02$ | $0.84\pm0.03$ |
| | | FCI-Heuristics | $21.30\pm3.50$ | $(61.60, 61.60)\pm(5.54, 5.54)$ | $0.41\pm0.18$ | $0.22\pm0.10$ | $0.29\pm0.13$ |
| | Data-LM$_{(t=0)}$ | LLM-first | $8.40\pm1.20$ | $(35.20, 35.20)\pm(2.92, 2.92)$ | $0.83\pm0.05$ | $0.69\pm0.02$ | $0.76\pm0.03$ |
| | | ILS-CSL | $8.50\pm2.60$ | $(30.25, 30.25)\pm(9.09, 9.09)$ | $0.83\pm0.08$ | $0.69\pm0.08$ | $0.75\pm0.07$ |
| | | BLANCE | $5.40\pm0.49$ | $(13.20, 13.20)\pm(2.40, 2.40)$ | $0.88\pm0.02$ | $0.83\pm0.02$ | $0.85\pm0.01$ |
| | Data-LM$_{(t=0.5)}$ | LLM-first | $9.60\pm1.69$ | $(38.70, 38.70)\pm(4.94, 4.94)$ | $0.80\pm0.06$ | $0.67\pm0.03$ | $0.73\pm0.05$ |
| | | ILS-CSL | $7.67\pm1.70$ | $(26.33, 26.33)\pm(3.68, 3.68)$ | $0.84\pm0.05$ | $0.74\pm0.04$ | $0.79\pm0.05$ |
| | | BLANCE | $4.60\pm0.49$ | $(12.00, 12.00)\pm(0.00, 0.00)$ | $0.91\pm0.03$ | $0.84\pm0.00$ | $0.87\pm0.01$ |
| USER LEVEL DATA-II ($d=8$) | Only-Data | FCI-Cumulative | $19.80\pm2.04$ | $(40.00, 40.00)\pm(3.03, 3.03)$ | $0.15\pm0.07$ | $0.10\pm0.04$ | $0.12\pm0.05$ |
| | | FCI-Vanilla | $17.40\pm1.83$ | $(36.80, 39.40)\pm(2.04, 3.38)$ | $0.07\pm0.13$ | $0.02\pm0.03$ | $0.03\pm0.05$ |
| | | FCI-Iterative | $20.60\pm1.77$ | $(42.80, 43.40)\pm(4.07, 4.22)$ | $0.06\pm0.08$ | $0.05\pm0.07$ | $0.05\pm0.07$ |
| | | FCI-Heuristics | $17.40\pm1.83$ | $(36.80, 39.40)\pm(2.04, 3.38)$ | $0.07\pm0.13$ | $0.02\pm0.03$ | $0.03\pm0.05$ |
| | Data-LM$_{(t=0)}$ | LLM-first | $18.33\pm0.94$ | $(42.67, 42.67)\pm(0.47, 0.47)$ | $0.21\pm0.04$ | $0.19\pm0.04$ | $0.20\pm0.04$ |
| | | ILS-CSL | $19.00\pm1.22$ | $(46.75, 46.75)\pm(4.87, 4.87)$ | $0.14\pm0.06$ | $0.13\pm0.07$ | $0.13\pm0.07$ |
| | | BLANCE | $18.67\pm1.70$ | $(41.00, 41.00)\pm(1.63, 1.63)$ | $0.22\pm0.02$ | $0.22\pm0.04$ | $0.22\pm0.02$ |
| | Data-LM$_{(t=0.5)}$ | LLM-first | $18.33\pm0.94$ | $(42.67, 42.67)\pm(0.47, 0.47)$ | $0.21\pm0.04$ | $0.19\pm0.04$ | $0.20\pm0.04$ |
| | | ILS-CSL | $17.60\pm2.33$ | $(41.20, 41.20)\pm(4.12, 4.12)$ | $0.21\pm0.11$ | $0.18\pm0.12$ | $0.19\pm0.12$ |
| | | BLANCE | $18.67\pm2.05$ | $(41.00, 41.00)\pm(1.63, 1.63)$ | $0.23\pm0.02$ | $0.22\pm0.04$ | $0.22\pm0.02$ |
| CHILD ($d=19$) | Only-Data | FCI-Cumulative | $27.50\pm0.00$ | $(111.00, 131.00)\pm(0.00, 0.00)$ | $0.38\pm0.00$ | $0.36\pm0.00$ | $0.37\pm0.00$ |
| | | FCI-Vanilla | $28.00\pm1.48$ | $(129.20, 133.20)\pm(10.46, 10.76)$ | $0.38\pm0.04$ | $0.26\pm0.05$ | $0.31\pm0.04$ |
| | | FCI-Iterative | $32.10\pm1.16$ | $(149.00, 164.40)\pm(7.16, 10.33)$ | $0.27\pm0.03$ | $0.26\pm0.04$ | $0.26\pm0.03$ |
| | | FCI-Heuristics | $28.00\pm1.48$ | $(129.20, 133.20)\pm(10.46, 10.76)$ | $0.38\pm0.04$ | $0.26\pm0.05$ | $0.31\pm0.04$ |
| | Data-LM$_{(t=0)}$ | LLM-first | $29.67\pm0.47$ | $(154.83, 154.83)\pm(9.75, 9.75)$ | $0.33\pm0.01$ | $0.35\pm0.02$ | $0.34\pm0.01$ |
| | | ILS-CSL | $29.80\pm2.48$ | $(160.00, 160.00)\pm(21.25, 21.25)$ | $0.33\pm0.05$ | $0.35\pm0.04$ | $0.34\pm0.04$ |
| | | BLANCE | $27.80\pm0.75$ | $(114.80, 114.80)\pm(7.76, 7.76)$ | $0.39\pm0.01$ | $0.45\pm0.03$ | $0.42\pm0.02$ |
| | Data-LM$_{(t=0.5)}$ | LLM-first | $30.67\pm1.11$ | $(146.83, 146.83)\pm(4.81, 4.81)$ | $0.31\pm0.03$ | $0.32\pm0.03$ | $0.32\pm0.03$ |
| | | ILS-CSL | $32.00\pm2.28$ | $(152.80, 152.80)\pm(12.95, 12.95)$ | $0.28\pm0.05$ | $0.29\pm0.06$ | $0.28\pm0.06$ |
| | | BLANCE | $28.10\pm0.86$ | $(100.60, 112.00)\pm(9.31, 0.00)$ | $0.40\pm0.01$ | $0.45\pm0.06$ | $0.42\pm0.01$ |
| ALARM ($d=37$) | Only-Data | FCI-Cumulative | $45.00\pm0.00$ | $(626.00, 626.00)\pm(0.00, 0.00)$ | $0.25\pm0.00$ | $0.02\pm0.00$ | $0.04\pm0.00$ |
| | | FCI-Vanilla | $49.50\pm1.61$ | $(617.80, 699.20)\pm(29.23, 68.43)$ | $0.00\pm0.00$ | $0.00\pm0.00$ | $0.00\pm0.00$ |
| | | FCI-Iterative | $52.40\pm6.21$ | $(612.20, 636.80)\pm(49.87, 43.83)$ | $0.33\pm0.14$ | $0.12\pm0.05$ | $0.17\pm0.07$ |
| | | FCI-Heuristics | $49.50\pm1.61$ | $(617.80, 699.20)\pm(29.23, 68.43)$ | $0.00\pm0.00$ | $0.00\pm0.00$ | $0.00\pm0.00$ |
| | Data-LM$_{(t=0)}$ | LLM-first | $57.33\pm1.89$ | $(695.33, 695.33)\pm(21.91, 21.91)$ | $0.22\pm0.03$ | $0.10\pm0.01$ | $0.13\pm0.01$ |
| | | ILS-CSL | $49.50\pm1.50$ | $(647.50, 647.50)\pm(18.94, 18.94)$ | $0.39\pm0.05$ | $0.10\pm0.02$ | $0.17\pm0.03$ |
| | | BLANCE | $51.60\pm1.77$ | $(595.00, 595.80)\pm(16.63, 16.94)$ | $0.40\pm0.04$ | $0.22\pm0.01$ | $0.28\pm0.01$ |
| | Data-LM$_{(t=0.5)}$ | LLM-first | $56.67\pm2.49$ | $(684.67, 684.67)\pm(16.34, 16.34)$ | $0.23\pm0.05$ | $0.10\pm0.02$ | $0.14\pm0.03$ |
| | | ILS-CSL | $49.75\pm1.48$ | $(633.25, 633.25)\pm(13.05, 13.05)$ | $0.41\pm0.02$ | $0.12\pm0.01$ | $0.19\pm0.01$ |
| | | BLANCE | $51.70\pm1.29$ | $(594.00, 595.20)\pm(38.75, 38.75)$ | $0.41\pm0.03$ | $0.22\pm0.02$ | $0.29\pm0.02$ |

performance by providing more accurate priors. As an evidence, we experiment with GPT-4o and GPT-5 model on EARTHQUAKE and ASIA dataset and showcase the superior performance (Modified SHD) in Table A2.

Table A4: **Ambiguity Compression in BLANCE:** We report the mean histogram entropy (mean and standard deviation over 5 runs) for the USER LEVEL DATA - I and USER LEVEL DATA - II datasets. A consistent reduction in entropy across batches indicates that BLANCE progressively compresses ambiguity.

| Dataset | Batch-1 | Batch-2 | Batch-3 | Batch-4 | Batch-5 | Batch-6 | Batch-7 |
|---|---|---|---|---|---|---|---|
| USER LEVEL DATA - I | $0.48 \pm 0.06$ | $0.50 \pm 0.05$ | $0.36 \pm 0.08$ | $0.17 \pm 0.13$ | $0.10 \pm 0.06$ | $0.12 \pm 0.06$ | $0.09 \pm 0.05$ |
| USER LEVEL DATA - II | $0.31 \pm 0.25$ | $0.27 \pm 0.22$ | $0.18 \pm 0.18$ | $0.12 \pm 0.19$ | $0.07 \pm 0.15$ | $0.00 \pm 0.00$ | $0.00 \pm 0.00$ |

**Ambiguity Compression**  In BLANCE, to explicitly account for uncertainty and ambiguity, we introduce a representational shift from DAGs to PAGs and therefore evaluate performance using Modified SHD and SID (Table 2), which are well-suited for uncertainty estimation in causal structures. In addition, we quantify ambiguity compression in BLANCE by reporting the mean histogram entropy across batches for the USER LEVEL DATA−I and USER LEVEL DATA−II datasets in Table A4. The decrease in histogram entropy over batches indicates that BLANCE progressively reduces structural ambiguity during the causal discovery process.