# OpenReview forum: "Language Models as Noisy Experts for Sequential Causal Discovery"
_ICLR.cc/2026/Conference — Submitted to ICLR 2026_

### Official Review · Reviewer_VGuz · 2025-10-20

**Soundness:** 3
**Presentation:** 3
**Contribution:** 2
**Rating:** 6
**Confidence:** 4

**Summary:**

The paper presents the BLANCE (Bayesian LM-Augmented Causal Estimation) framework for causal structure discovery and parameter estimation in settings where data arrives in sequential, batch-wise fashion. Traditional causal discovery methods typically assume access to complete data and the availability of reliable domain experts, but in practice, data often arrives in batches, is subject to sampling bias, and expert knowledge is scarce. BLANCE treats Language Models (LMs) as noisy surrogate experts to address both data-induced bias and LM-induced bias (such as hallucinations and inconsistency).

**Strengths:**

1. The paper makes an innovative and systematic contribution by identifying and resolving the dual biases inherent in combining LMs with sequential data: namely, data-induced bias (from issues like sampling bias in batches) and LM-induced bias (stemming from factors like hallucination and inconsistency). The BLANCE framework provides a hybrid Bayesian solution to systematically integrate and mitigate both sources of error.

2. The experimental results are presented comprehensively and logically. The use of clear visualizations and metrics effectively conveys the purpose and efficacy of different experiments, allowing for a clear understanding of the framework's performance evolution during sequential learning and the specific contributions of key components (like sequential optimization and the dynamic threshold).

3. The paper demonstrates a strong commitment to scientific reproducibility. By providing extensive experimental details, hyperparameter configurations, LM prompt templates, and comprehensive descriptions of the datasets and simulation processes in the appendix , the authors significantly enhance the credibility and reproducibility of their findings.

**Weaknesses:**

1. The BLANCE framework, while effectively integrating LM knowledge into sequential data in an engineering sense, lacks depth in the foundational innovation of its methodology. Firstly, the framework adopts the PAG structure (implemented via the FCI algorithm) to handle uncertainty arising from latent confounders and selection bias , but this is a standard and mature theoretical tool in causal discovery theory for addressing the challenge of Causal Insufficiency, and is not original to BLANCE. Secondly, its Bayesian approach, which converts LM knowledge into a prior to guide structure learning, also appears to be a direct application of existing ideas for integrating domain knowledge within Bayesian causal discovery.
2. Although Bayesian parameter estimation aims to address the challenge of latent confounders, its underlying assumption is the Linear Gaussian SEM. This limitation may not be applicable to many real-world scenarios featuring non-linear or non-Gaussian noise, thereby restricting the framework's general applicability.
3. While the LM is queried to identify confounder variables , this process is itself heuristic and relies on the LM's world knowledge, offering no theoretical guarantee that the correct latent variable will be found. If the LM fails to identify the correct latent variable, the subsequent parameter estimation will be based on a flawed premise.

**Questions:**

1. How is the hyperparameter $\alpha$ in the dynamic background knowledge threshold $\tau_i^e$ (Equation (5)), which balances the posterior entropy uncertainty and sampling uncertainty, systematically chosen by the paper? Does the optimal value of $\alpha$ remain consistent across different datasets or varying LM noise levels? If sensitive, can a mechanism for adaptively adjusting $\alpha$ be provided?

2. To enhance the general applicability of the framework, it is suggested to extend the Bayesian parameter estimation to non-linear/non-Gaussian models. For example, by using flow-based or Variational Inference methods to handle non-linear SEMs, while retaining the ability to integrate LM priors. This would enable BLANCE to address a broader range of practical problems.

3. Currently, the framework treats all LM responses as equally weighted. Considering that LMs can express confidence in their judgments in their output, this confidence could be included in the construction of the histogram, rather than just using a binary indicator function. This would provide more fine-grained modeling of LM noise.

4. It could be considered to combine the sequential optimization of LM interaction with active learning strategies. Maximize information gain by querying the results of interventions or conditional interventions.

---

> ### Author Response · Authors · 2025-11-21
>
> We thank the reviewer for their thoughtful comments. We appreciate the strengths of our work that you have noted. We address both questions and weaknesses below.  Recall BLANCE is our method.
>
> **(Q1) How is the hyperparameter $\alpha$ in the dynamic background knowledge threshold? If sensitive, can a mechanism for adaptively adjusting be provided?**
> The hyerparameter $\alpha$ in the dynamic background knowledge threshold Eq(5) weights two sources of uncertainty *i.e.* posterior entropy (uncertainty in the histogram edge distribution), and the sampling uncertainty (which decreases as more LM interactions accumulate). By design, $\alpha$ allows flexible weighting between these two sources depending on dataset characteristics. For experiments, we perform a grid search and choose the $\alpha$ value. As mentioned in the Appendix G.6, the $\alpha$ equals 0.5 for User-Level Data II and 0.3 for the other datasets.
>
> **(Q2 and W2) Linear SEM assumption to real-world applications.**
> In the current formulation, the linear structural equation model (SEM) assumption plays a role in the parameter estimation via the EM-style updates over the latent confounder. Relaxing this assumption introduces technical challenges, particularly in the E-step, where computing the exact posterior over the latent variable becomes intractable in the non-linear case. A plausible direction to overcome this is to use approximate inference techniques, such as variational inference, to estimate the posterior. While the core structure of the BLANCE algorithm remains intact, key components would need to be approximated in a principled manner. As rightly pointed out, Variational Inference methods can be used to handle non-linear SEMs which we also discuss in the future work section.
>
> **(Q3) Confidence from language models to specify uncertainty in the response?** This point, relating to causal structure discovery, is intuitively appealing. LM outputs are typically deterministic or probabilistic over tokens, but do not natively produce well-calibrated confidence scores[1]. To overcome this obstacle, we model LM responses as noisy samples aggregated over multiple calls, which allows empirical estimation of uncertainty through histogram-based posteriors. This repeated sampling approach empirically reduces hallucination effects and captures the LM’s evolving beliefs more robustly than a single-shot confidence estimate, even if LM could return well-calibrated confidence, which LMs do not. A survey [2] brings out the unreliability of confidence scores, and LM-confidence is an active area of research with increasing interest.
>
> [1] Rui Li, Jing Long, Muge Qi, Heming Xia, Lei Sha, Peiyi Wang, and Zhifang Sui. Towards Harmonized Uncertainty Estimation for Large Language Models. In Proceedings of Association for Computational Linguistics, 2025
>
> [2] Geng, Jiahui, et al. "A survey of confidence estimation and calibration in large language models." Proceedings of the 2024 Conference of the North American Chapter of the Association for Computational Linguistics: Human Language Technologies (Volume 1: Long Papers). 2024.

---

> > ### Author Response · Authors · 2025-11-21
> >
> > **(Q4) Maximize information gain by querying the results of interventions or conditional interventions**
> > We appreciate this suggestion. Active learning combined with interventional queries is indeed a promising direction. The intuition that interventions directly reveal causal effects, which could provide richer information along with observational data seems correct. Currently, BLANCE is designed primarily for the observational causal discovery setting. However, extending BLANCE to incorporate interventional strategies and active learning would be a valuable future direction, which we will note in the paper.
> >
> > **(W1) Lacks depth in the foundational innovation of its methodology**
> > We shift from DAG to PAG for the reason stated in point one of the review and to address causal insufficiency. However, in a difference with prior art which is situated in having access to the full dataset, our sequential, batched data - a reality of digital interactions data - makes the problem of updating the causal structure, as new data come in, *novel*. This throws new challenges too. About the second point, Bayesian updating is indeed an established concept. The novelty herein arises in its application to updating prior information as new batch of data arrives and eliciting information from an LM, iteratively.
> >
> >
> > **(W3) What if the latent confounder predicted is wrong?**
> > We acknowledge that latent confounder identification is a heuristic process dependent on LM world knowledge. However, we note two important points: (1) In practice, LMs tend to suggest semantically plausible confounders. For example, in the Red Wine Quality dataset (Figure 5), the LM correctly identified alcohol as the primary confounder between density and quality, aligning with domain knowledge. (2) Parameter estimation is robust to confounder misspecification. Figure 6 demonstrates that even with severely misspecified prior distributions (*e.g.,* $N(50, 1.5)$ vs. true $N(11, 1.0)$), the Bayesian EM algorithm converges to correct parameter estimates as more batches of data accumulate. This robustness arises because the likelihood term in Eq. (11) is data-driven; incorrect LM priors act as weak regularization progressively overcome by evidence. We acknowledge that hallucinations can occur, potentially leading to $L_2$ estimation error if the LM identifies an incorrect confounder. Systematic evaluation of LM accuracy in confounder identification across diverse domains remains important future work.
> >
> > We have updated the manuscript with new LM model experiments (Table 5) as suggested by reviewers. We have updated the future work section. Please let us know if you have any further questions or suggestions.

---

### Official Review · Reviewer_nVWr · 2025-10-31

**Soundness:** 3
**Presentation:** 3
**Contribution:** 3
**Rating:** 6
**Confidence:** 3

**Summary:**

This work presents a new method for merging background causal relation knowledge extracted from LLMs with traditional data-based causal discovery algorithms. The method consists in iterating between two steps: one that uses a data sample and causal discovery algorithm like FCI to learn a Partial Ancestral Graph (PAG) representation (which contains bidirected edges representing latent confounder as well as 'undecided' endpoints that represent ambiguity related to edge direction) and another one where the LLM is queried about the direction of ambiguous edges to refine. The two steps are iterated in a sequential Bayesian updating fashion to derive a posterior distribution over PAGs from a current prior distribution and data likelihood distribution. Given the possibility of latent variables, the latter step is performed by Bayesian inference, assuming the local models are linear Gaussian. Empirical results with simple benchmark DAGs as well as datasets from competitions show that the proposed method achieves better reconstruction metrics than data-driven methods alone and other use of LLM-backed causal discovery.

**Strengths:**

The paper is overall well-written, the topic is relevant and the contribution is novel. The work is for the most part well described, except from some techniques (e.g.LLM pairwise or triplet prompting). The empirical results are promising, although only show experiments with small graphs.

**Weaknesses:**

The paper is mostly an empirical approach, lacking theoretical justification. At the same time, the method requires many additional  assumptions. The authors do not discuss to which extent one can assume such assumptions in real world scenarios. The benchmarks used are very common, which means LLMs introduce a lot of bias towards the correct structure. The exception is the User Level Data datasets, where a learned DAG (using the whole dataset) is used as ground truth. The graphs are also relatively small, which makes guessing easier.

A common pitfall of the use of LLM as experts is that their background knowledge might contradict a given dataset domain. The benchmarks used do not allow for testing this effect.

The proposed method has many moving parts: the LLM prior extraction as histograms, the refinement of a PAG, the use of a PAGs, the casting of the problem as Bayesian inference, the Bayesian parameter inference, and the streamed data. This makes it difficult to assess the contribution of the separated parts, and also casts out possible competitors. I would be more confident about the empirical results if those moving parts were evaluated independently, especially if the stream data was ignored at first. In fact, it is not very clear how common is this situation, where data arrives in batches and we need to iteratively update a causal dag from it.

**Questions:**

I could not understand how the posterior distribution over PAGs is represented. The expert knowledge extracted from the LLM is represented as histograms over edges. How is this is combined with the likelihood to obtain a posterior? Do you assume that the posterior factorizes as edge distributions as well?

---

> ### Author Response · Authors · 2025-11-21
>
> We thank the reviewer for their thoughtful comments. We appreciate the strengths of our work that you have noted. We address both questions and weaknesses below.  Recall BLANCE is our method.
>
> * **Many additional assumptions .... in real-world scenarios.**
> Allow us to address your concern by referring to Section 2 of the paper, where we *explicitly detail* all the assumptions. Assumptions 1, 2, 5 are standard in the extant prior art. Assumption 3 is actually more realistic than extant prior art since we *allow for* selection bias and confounding variables; and *relax* Faithfulness, Causal Sufficiency, and Population Inference assumptions found in much of prior art. Assumption 4 is very realistic since companies hold data only for a limited look back window. Assumptions 6 and 7 about single confounder per variable and atomic confounding are related and not unusual in the literature. Assumptions 6 and 7 aid in identification. Considering these assumptions, overall, they offer a good trade-off between ecological validity with regarad to real world scenarios and the research literature.
>
> * **The benchmarks used are very common ... LLMs introduce a lot of bias towards the correct structure.**
> Presumably by benchmarks the reviewer meant the datasets used. The baseline models against which we compare BLANCE also use LLMs, making the comparison with those baselines like-to-like. Yet, on both these benchmark datasets and the new User Level Dataset - 1, our approach comfortably outperforms all baselines. In this rebuttal, we now add results from additional LMs - QWEN and Llama3.1 - as well as one other recent model as baseline (BFS). Across all these variety of evaluations, BLANCE shows robustness and performance better than all baselines and across benchmark datasets.
> One implication of the reviewer's comment is the potential issue of memorization of datasets seen by LMs. Please read our response **(W8) memorization by LLM** to reviewer **yAbn**.
>
> * **Graphs are also relatively small making guessing easier**
> We acknowledge that causal discovery on smaller graphs may be relatively easier. However, the fundamental challenge in causal discovery—especially in hybrid settings with noisy expert knowledge is not primarily driven by graph size, but by the quality and consistency of the observational data and the reliability of expert responses, we term them as *data-bias* and *LM-bias*, respectively. BLANCE's contribution is in principally integrating these two sources under dual biases, a problem that remains challenging regardless of graph dimensionality.
>
> * **LLM as experts pitfall is that their background knowledge might contradict a given dataset domain.**
> We agree, and this is precisely the motivation for BLANCE. Only-LM methods are indeed more prone to domain misalignment. BLANCE is a hybrid, *data-first* framework, where we specifically admit LM as a *noisy* expert. The LM provides a prior for the causal discovery algorithm, but critically, when LM knowledge contradicts observed data patterns, the Bayesian formulation (Eq. 3) naturally downweights LM suggestions, allowing the data likelihood to dominate in domain-specific settings. This ensures BLANCE can handle scenarios where LM background knowledge is misaligned or absent.
>
> * **The proposed method has many moving parts ......... moving parts were evaluated independently.**
> We acknowledge about the multiple parts in BLANCE. This comes with the territory of causal discovery in a sequential batched data setting, the problem which BLANCE tackles. BLANCE is designed for sequential causal discovery with streaming batch data, which is the primary focus of the work. Table 2 showcases the motivation for using PAG, under the non-streaming scenario. Ablations provide a way to evaluate the parts separately. We provide ablations of the key components of BLANCE i.e. proposed selection score vs. random selection in Figure 4. as well as provide ablations on the fixed vs. dynamic background knowledge threshold in Figure 4. We discuss them in Section 5.1.
>
> * **Motivation about the sequential batch settings.**
> One of the largest contributors of data are digital interactions that all users indulge in everyday with sites and apps. Every digital company collects and uses the data and a large number of decisions are made from them. The data are processed at periodic time interval, often weekly, which maps directly to our sequential batch setting. The decisions made thereupon benefit from causal structure discovery since effect of actions on outcomes is of great interest to the companies.

---

> > ### Author Response · Authors · 2025-11-21
> >
> > * **I could not understand how the posterior distribution over PAGs is represented. The expert knowledge extracted from the LLM is represented as histograms over edges. How is this combined with the likelihood to obtain a posterior? Do you assume that the posterior factorizes as edge distributions as well?**
> > We would like to clarify that BLANCE does not maintain an explicit posterior distribution over the entire space of PAGs. Instead, it represents the posterior implicitly by combining statistical evidence and LM-derived knowledge within the constraint-based causal discovery performed by FCI algorithm. LM provides edge-wise pseudo priors in the form of empirical histograms, and these are converted into background knowledge only when the pass the dynamic background threshold value, Eq. (5). At each batch, FCI processes the observational data under these accumulated constraints, so the likelihood term is expressed through conditional independence tests, while the LM prior puts constraints on certain causal relationships (edges in the causal structure). Thus, BLANCE integrates LM knowledge and data in a Bayesian-update sense while avoiding the intractability of maintaining a full posterior over PAGs.
> >
> > We have updated the manuscript with new LM model experiments (Table 5) as suggested by reviewers. We have updated the future work section. Please let us know if you have any further questions or suggestions.

---

### Official Review · Reviewer_yAbn · 2025-11-01

**Soundness:** 2
**Presentation:** 3
**Contribution:** 2
**Rating:** 2
**Confidence:** 4

**Summary:**

The paper focuses on causal discovery with an emphasis on a setting where we get sequential batches of observational data. It makes sense to tackle data bias and LLM induced biases, but the data sampling method does not seem to completely address this issue as it is still from the same distribution. Secondly the benchmarks the paper focuses on are standard benchmarks which have been found to be memorized by LLMs in recent studies. Benchmarking GPT-3.5-Turbo for majority of the experiments which is an outdated model does not provide extensive understanding of how effective their pipeline is. Although the authors did use GPT-4o and GPT-5 for two datasets, they don't tackle the claims of memorization which can definitely skew their results as shown by recent literature. Some assumptions are very strong making the setting too simple, and would not hold in realistic setting. Overall I think the framework proposed is promising, but lack of extensive empirical evaluations and strong assumptions make it difficult to understand its effectiveness, especially in a realistic setting.

**Strengths:**

1. This framework addresses a real gap in causal discovery, going beyond the static datasets used by prior work. Sequential (batch-wise) data arrival, where each batch can be biased or incomplete is a realistic problem that is underexplored.

2. I like the shift from DAGs to PAGs to model uncertainty and ambiguity.

3. Optimizing LLM calls via sequential optimization is a good idea to constrain the high inference cost of LLMs.

**Weaknesses:**

Some weaknesses I would like the authors to tackle:

1. BLANCE assumes a linear Gaussian SEM for all causal relations (Assumption 5). This heavily restricts its applicability to real-world, nonlinear systems (e.g., biological, economic, or text-based phenomena). Many causal discovery tasks today rely on nonlinear additive noise models (ANMs) or neural causal discovery, so results may not generalize.

2. The work assumes the true causal structure remains the same across batches which is a very big assumption. In reality, sequential or streaming data can easily violate this assumption [1].

3. No clear understanding of how much ambiguity is compressed across samples by the proposed framework.

4. In BLANCE, the histograms are deterministically updated meaning there’s no principled measure of how uncertain those priors are beyond empirical entropy. The “LM-derived priors” in BLANCE are empirical frequency tables that record how often the LM predicted each causal relation type. They’re called priors because they influence the next batch’s belief, but mathematically they’re pseudo-counts, not fully parameterized or normalized probability distributions. Please correct me if I am wrong here.

5. It would be helpful to see how well the framework performs beyond using FCI to initialize PAGs. If the authors can provide some results for PC, Lingam, SCORE, CamML or other algorithms, to show the generalization of their framework, that can add to their framework's impact.

6. While the final metrics focus only on SHD, or final graph based edges, no analysis of ambiguity or how LLM induced bias is reduced by their framework is done. Since these were the main points targetted by the authors, I would appreciate some analysis about that.

7. They compare against ILS-CSL (2023) and LLM-First baselines, but omit recent hybrid approaches like [2]

8. Benchmarking GPT-3.5-Turbo for majority of the experiments which is an outdated model does not provide extensive understanding of how effective their pipeline is. Although the authors did use GPT-4o and GPT-5 for two datasets, they don't tackle the claims of memorization which can definitely skew their results as shown by recent literature [3]. I understand the issue with high cost required for running inference for these models, but evaluating on a suite of open models like Llama, Qwen, Phi could have been explored.

9. BNLearn is outdated for LLM based evaluation due to high chances of data contamination which are confirmed by [3]. The paper's results would have been stronger if they evaluated on recent benchmarks, or benchmarks released after the training cut off of the model.

Overall I think the work is promising, but lacking in its empirical and theoretical grounding.

[1] Causal Discovery for Non-stationary Non-linear Time-series Data Using Just-In-Time Modeling. Fujiwara et al. 2023
[2] Efficient Causal Graph Discovery Using Large Language Models Bengio et al. 2024
[3] Realizing LLMs’ Causal Potential Requires Science-Grounded, Novel Benchmarks Srivastava et al. 2025

**Questions:**

Please refer the weaknesses section, and clarify the points. I am happy to increase my score if the authors can provide justification and empirical results to tackle these points, especially the memorization issue.

---

> ### Author Response · Authors · 2025-11-21
>
> We thank the reviewer for their thoughtful comments. We appreciate the strengths of our work that you have noted. We address both questions and weaknesses below. Recall BLANCE is our method.
>
> **(W1) SEM assumption is too strong:** It is worth clarifying that the linear structure equation model (SEM) assumption is not used in the causal structure discovery. Thus, the concern about generalization to nonlinearity does not apply to our Bayesian causal discovery part. Linear SEM is assumed only in the Bayesian parameter estimation via the EM-style updates over the latent confounder. Relaxing this assumption introduces technical challenges, particularly in the E-step, where computing the exact posterior over the latent variable becomes intractable in the non-linear case. A plausible direction to overcome this is to use approximate inference techniques, such as variational inference, to estimate the posterior. While the core structure of the BLANCE algorithm remains intact, key components would need to be approximated in a principled manner and is an avenue for future work in parameter estimation.
>
> **(W2) Assumption that true causal structure is stationary:** Whether real world streaming or sequential data are stationary or non-stationary is an empirical question. It is hard to say one way or another across all settings which generate such data. The types of batched data emanating from clickstream in websites and apps, a motivation for our work, have patterns which are quite regular, over hours of the day, days of the week, weeks of the month, excluding holiday season's main effect. Thank you for referring us to the paper [1] Fujiwara et al 2023. While this paper offers a model for non-stationary, non-linear time series data, when it comes to empirical demonstration they use synthetic data for all their experiments, but avoids real world data. All things considered, our assumption of stationarity seems reasonable for some batched data setting.
>
> **(W3 and W6) Analysis of ambiguity:** While we report SHD, SID, Precision, Recall, and F1 Score which are standard causal discovery metrics in prior art, we agree that LM-bias is not explicitly quantified currently in the manuscript. We assume that these effects are implicitly captured in the reported metrics in terms of performance, as LM-only methods consistently perform worse than BLANCE across all metrics.
>
> **(W4) Mathematically histograms are pseudo-counts and not fully parameterized priors:** We appreciate the reviewer’s observation and would agree that BLANCE uses LM-derived frequency histograms as empirical pseudo-priors rather than fully parameterized Bayesian priors. This is a design choice we took as LMs can exhibit non-stationary and context-dependent error patterns. Due to this, we found that maintaining deterministic pseudo-counts over relation types provided a stable and interpretable mechanism for incorporating LM noisy responses across batches without assuming a specific generative model for LM uncertainty. These pseudo-priors still function as prior information in the sequential Bayesian update by guiding and influencing posterior structure while avoiding the overhead and assumptions required to model LM reliability explicitly. We discuss in future work how a Dirichlet or hierarchical Bayesian prior over LM judgments could serve as a natural extension.

---

> > ### Comment · Reviewer_yAbn · 2025-11-27
> > **Response to Rebuttal**
> >
> > I thank the authors for their rebuttal.
> >
> > > We assume that these effects are implicitly captured in the reported metrics in terms of performance, as LM-only methods consistently perform worse than BLANCE across all metrics.
> >
> > I believe this is a weak argument, since the paper makes a strong emphasis about their method capturing ambiguity, but don't provide any analysis to support this claim. Improvement in F1 scores is a proxy, but some qualitative or quantitative analysis would have been appreciated given the strong emphasis on tackling ambiguity.
> >
> > > All things considered, our assumption of stationarity seems reasonable for some batched data setting
> >
> > I disagree, the authors claim that clickstream data in website and apps are regular in nature, and therefore it doesn't matter if the data is stationary or over a period of time since this type of data is the motivation of their work. However majority of the datasets that they have tested are from BNLearn repositories which are healthcare datasets. Also, claiming that time series data is similar to stationary data without any fundamental guarantee is a weak argument given that one of the main contribution of the work that tries to set themselves apart from the extensive prior literature focusing on using LLMs for causal discovery is how their method works well for temporal data. It would have been more convincing if the authors had done some experiment (even on synthetic data) as depicted by [1] Fujiwara et al 2023 to capture the true essence of time series data.
> >
> > Thank you for providing clarification on the remaining points.

---

> ### Author Response · Authors · 2025-11-21
>
> **(W5) Other causal discovery algorithms:** We focus on a setting where *both* selection bias and latent confounding are present (*cf.* Assumption 3). To accommodate both aspects in this general setting, it is necessary to shift from DAGs to Partial Ancestral Graphs (PAGs), since this explicitly captures uncertainty due to latent confounders and selection bias. The Fast Causal Inference (FCI) algorithm naturally aligns with these assumptions, making it a principled choice for our framework.
>
> Moreover, our goal is not to benchmark causal structure discovery algorithms, but rather to demonstrate how BLANCE can refine causal structure over time by integrating noisy expert knowledge (from LMs) with observational data. Notably, BLANCE is designed to be algorithm-agnostic, and can be used in a plug-and-play manner with any PAG-generating causal discovery method.
> Below, we briefly discuss why several modern algorithms are not directly applicable:
> - **Greedy Equivalence Search (GES):** GES assumes both causal sufficiency and the absence of selection bias. These assumptions do not hold in our sequential setting, where both latent confounders and selection bias may be present.
> - **DAG-NoCURL and DAG-NoTEARS:** These are continuous optimization-based methods that output DAGs under the assumption of causal sufficiency, without accounting for latent confounders. Thus, they do not apply to our setting, which requires a PAG-based representation to model uncertainty.
> - **DAGMA:** DAGMA is a score-based causal discovery method that also formulates the problem as a continuous optimization task. Like DAG-NoCURL and DAG-NoTEARS, it outputs deterministic DAGs and assumes no latent confounding, making it incompatible with the sequential setting.
> - **LiNGAM:** LiNGAM (Linear Non-Gaussian Acyclic Model) assumes that all relevant variables are observed, *i.e.*, there are no hidden confounders. This causal sufficiency assumption renders it unsuitable for the sequential setting.
> - **Recursive Causal Discovery (RCD):** Unlike FCI, RCD produces a DAG rather than a PAG, focusing on efficient DAG learning. Hence, when dealing with latent confounders and selection bias requiring PAG representation, RCD can not be employed.
>
> We note that enhanced variants of FCI such as RFCI, GFCI, and FCI+ are fully compatible with BLANCE and can be seamlessly integrated bringing their advantage to BLANCE as well. We keep the research to study convergence impact of different algorithms for future work.
>
> **(W7) Omit recent hybrid approaches like [2]:** Thank you for pointing us to Bengio and co-authors' paper [2], Efficient Causal Graph Discovery Using Large Language Models Jiralerspong et al. 2024 (hereafter, BFS); we inadvertantly missed it. We now compare BLANCE against the BFS method on these causal discovery datasets. We observe that while the BFS method relies primarily on LM without grounding to data, BLANCE's data-first hybrid approach—which iteratively grounds LM suggestions in local observational data—achieves stronger performance on datasets less likely to be memorized.
> We use a recent model **GPT4.1-nano (2025-04-14)** and evaluate on two datasets. We particularly choose GPT4 family as in their paper they used the GPT4 family for experiments. First, we experiment with a mid-size dataset from BNLearn repository *Child* and a real-world Google *User Level Data - I*. Below we report mean and standard deviation over 5 runs. The results clearly favor BLANCE over BFS.
>
> |Dataset| Method | Mod. SHD | SID | Precision | Recall | F1 Score|
> | ---| --- | -------- | ---- | -------- | ---- | -------- |
> |User Level Data-I| BFS | 36.6 ± 8.96 | (65.8, 65.8) ± (2.17, 2.17) | 0.13 ± 0.06 | 0.29 ± 0.14 | 0.18 ± 0.06 |
> || BFS with correlation | 24.0 ± 6.24 | (42.2, 42.2) ± (6.53, 6.53) | 0.17 ± 0.12 | 0.28 ± 0.11 | 0.21 ± 0.11 |
> ||BLANCE| **10.6 ± 2.13** | **(34.8, 35.2) ± (5.56, 5.84)** | **0.97 ±0.04** | **0.43 ± 0.06** | **0.60 ± 0.06** |
> |Child| BFS | 34.75 ± 7.80 | (252.50, 252.5) ± (23.36, 23.36) | 0.18 ± 0.08 | 0.12 ± 0.04 | 0.14 ± 0.03 |
> || BFS with correlation | 32.3 ± 4.16 | (301.1, 301.1) ± (14.42, 14.42) | 0.21 ± 0.03 | 0.15 ± 0.01 | 0.18 ± 0.01 |
> ||BLANCE| **27.90 ± 1.61** | **(143.5, 165.5) ± (12.23, 35.97)** | **0.42 ±0.06** | **0.23 ± 0.04** | **0.30 ± 0.03** |

---

> ### Author Response · Authors · 2025-11-21
>
> **(W8) Outdated model ...... but evaluating on a suite of open models like Llama, Qwen, Phi could have been explored:** We now evaluate BLANCE across two additional, diverse LLM families **Qwen and Llama**, on top of GPT whose results are in the paper. The QWEN and Llama based results are remarkably consistent with that of GPT, demonstrating robustness across models which have different training data, architectures and memorization profiles.
>
> We experiment with two open source LMs, Llama3.1(8B-instruct) and Qwen3(4B-instruct), on two datasets. First, a mid-size dataset from BNLearn repository *Child* and a real-world Google *User Level Data - I*. Below we report mean and standard deviation over 5 runs. The results show that BLANCE outperforms the benchmarks with these LMs as well, supporting our thesis.
>
> |Dataset | Model | Method | Mod. SHD | SID | Precision | Recall | F1 Score|
> |----| -------- | --- | -------- | ---- | -------- | ---- | -------- |
> |User Level Data-I|Qwen3-4B-instruct| LLM-first  | 23.67 ± 1.11 | (64.33, 64.33) ± (0.94, 0.94) | 0.35 ± 0.04  | 0.28 ± 0.04  | 0.31 ± 0.04  |
> ||| ILS-CSL    | 20.60 ± 4.18 | (56.80, 56.80) ± (7.36, 7.36) | 0.45 ± 0.13  | 0.36 ± 0.10  | 0.40 ± 0.12  |
> ||| BLANCE     | **9.8 ± 2.04**   | **(26.6, 26.6) ± (6.83, 6.83)**   | **0.91 ± 0.08**  | **0.54 ± 0.08**  | **0.67 ± 0.07**  |
> ||Llama3.1-8B-instruct| LLM-first  | 11.5 ± 0.5   | (42.5, 42.5) ± (1.5, 1.5) | 0.74 ± 0.01   | 0.61 ± 0.03   | 0.67 ± 0.02   |
> ||| ILS-CSL    | 9.75 ± 1.64  | (32.25, 32.25) ± (3.56, 3.56) | 0.78 ± 0.05   | 0.66 ± 0.03   | 0.72 ± 0.04   |
> ||| BLANCE     | **7.5 ± 0.45**   | **(25.8, 25.4) ± (2.32, 4.22)**   | **0.92 ± 0.07**   | **0.69 ± 0.01**   | **0.79 ± 0.01**   |
> |Child|Qwen3-4B-instruct| LLM-first  | 35.80 ± 2.04 | (173.8, 173.8) ± (20.82, 20.82) | 0.27 ± 0.02  | **0.42 ± 0.02**  | 0.34 ± 0.01  |
> ||| ILS-CSL    | 30.60 ± 2.42 | (196.80, 196.80) ± (14.13, 14.13) | 0.30 ± 0.06  | 0.28 ± 0.05  | 0.29 ± 0.05  |
> ||| BLANCE     | **21.2 ± 0.75**   | **(107.8, 107.8) ± (13.42, 13.42)**   | **0.51 ± 0.01**  | 0.32 ± 0.03  | **0.39 ± 0.02**  |
> ||Llama3.1-8B-instruct| LLM-first  | 45.80 ± 1.94 | (211.2, 211.2) ± (15.25, 15.25) | 0.13 ± 0.01  | 0.19 ± 0.02  | 0.16 ± 0.01  |
> ||| ILS-CSL    | 27.60 ± 2.24 | (146.40, 146.40) ± (15.93, 15.93) | 0.37 ± 0.05  | 0.38 ± 0.07  | 0.38 ± 0.06  |
> ||| BLANCE     | **24.6 ± 0.80**   | **(101.8, 101.8) ± (8.11, 8.11)**   | **0.44 ± 0.01**  | **0.45 ± 0.02**  | **0.44 ± 0.01**  |

---

> > ### Author Response · Authors · 2025-11-21
> >
> > **(W8) LLM Memorization:** We acknowledge the potential issue of memorization by LM on canonical causal discovery benchmarks and thank you for referring us to [3]. To address this, we do not rely only on standard benchmark datasets. Instead, we conduct additional experiments on two **real world** Google User Level Datasets, which, to our best knowledge, have not been previously used for causal discovery and thus any causal structure out of this data is unlikely to have been seen by an LM. As way of additional information, the actual large dataset (referred in paper and being reproduced here https://www.kaggle.com/datasets/bigquery/google-analytics-sample/data) is raw, time-stamped, clickstream data and the column headers therein are different from the attributes used in our two user datasets. The column headers are *visitorId, visitNumber, visitId, visitStartTime, date, totals, trafficSource, device, geoNetwork, customDimensions, hits, fullVisitorId, userId, channelGrouping, socialEngagementType*. Crucially, for our dataset, we curated attributes using rules applied to a randomly sampled portion of the dataset, which was itself preprocessed from the raw data. The column headers do not have a direct semantic mapping to attributes we use. Even if an LM has seen the column headers from the publicly available data, it is unlikely that the attributes we use were seen, let alone any causal structure  out of them is seen.
> > To check for memorization we now run experiments on these real world datasets to provide some empirical evidence. Refer to the first Table above. The *Child* dataset's causal graph is likely to have been seen by GPT, since prior art has used this causal graph. The *User Level Data - I* dataset's causal graph is unlikely to have been seen by any LM, as explained above. First, the results show that with either dataset BLANCE outperforms BFS. Let us investigate the difference in differences. Second, for the *Child* dataset which is likely seen by the LM, the improvement BLANCE shows over BFS is **smaller** than the improvement BLANCE shows over BFS for the *User Level Data - I*, which is unlikely seen by the LM. Third, the performance of BFS is worse in *User Level data - I* than in *Child*, suggesting BFS's dependence on dataset and causal graph seen by the LM. This is consistent with our reading of the paper's limitation in Sec 5: "First, since LLMs rely on the knowledge from their training data to synthesize the causal relationships, the proposed approach only works on real-world data." Our approach outperforms BFS with and without dependence on data and causal graph seen by LM.
> >
> > We have updated the manuscript with new LM model experiments (Table 5) as suggested by reviewers. We have updated the future work section. Please let us know if you have any further questions or suggestions.

---

> > > ### Comment · Reviewer_yAbn · 2025-11-27
> > > **Followup to Rebuttal**
> > >
> > > > Memorization Claim
> > >
> > > I thank the authors for running this experiment. I am not sure if I am convinced with this ablation completely. A concern is the way “ground truth” is constructed for the USER datasets. The authors first run DirectLiNGAM on the full observational data and then treat the resulting graph as the true causal structure for evaluation. However, DirectLiNGAM is itself a causal discovery algorithm with strong modelling assumptions (linearity, non-Gaussian noise, causal sufficiency, acyclicity). If these assumptions are even moderately violated, the estimated graph can be systematically wrong, so using it as ground truth effectively evaluates how well other methods recover DirectLiNGAM’s inductive biases rather than the underlying data-generating process. This is quite different in spirit from standard benchmarks (e.g., many bnlearn networks), where the graph is expert-designed or otherwise curated and then data is sampled from that known structure. Here, the same dataset is used both to infer a model and then to elevate that model to an oracle, which risks baking in model misspecification and sample idiosyncrasies. At minimum, the authors should acknowledge this limitation and, ideally, either (i) evaluate on datasets with genuinely known or expert-validated causal graphs, or (ii) treat DirectLiNGAM’s output as one candidate structure and probe robustness to alternative estimates rather than presenting it as ground truth.

---

> > > > ### Author Response · Authors · 2025-11-30
> > > >
> > > > We are glad that our rebuttal addressed the reviewer's concerns. The reviewer has raised three points in the response, which we now address.
> > > >
> > > > **Ambiguity:** Firstly, the representation shift from DAG to PAG proposed in BLANCE provides a principled way to represent uncertainty in the causal structure. The benefit of this alone is visible in Table 2, which depicts the benefit of PAG to represent inherent causal uncertainty. In this regard, we draw the reviewer's attention to the two metrics used, **SID** and **Mod. SHD**. These are used specifically to measure the uncertainty in the output causal structure, where arrows with uncertainty, such as o-> vs. ->, are treated differently (discussed in Appendix Section D).
> > > >
> > > > To directly measure ambiguity compression, we now report the **mean histogram entropy** for each batch (mean ± stddev) over 5 runs in the Table below. The *decrease* in entropy of the histogram over batches shows the reduction in ambiguity. We have also added this to the manuscript (Table A4).
> > > >
> > > > |Dataset | Batch 1 |Batch 2 |Batch 3 |Batch 4 |Batch 5 |Batch 6|Batch 7 |
> > > > |--|--|--|--|--|--|--|--|
> > > > |User Level Data - I| 0.48 ± 0.06| 0.50 ± 0.05| 0.36 ± 0.08| 0.17 ± 0.13 | 0.10 ± 0.06| 0.12 ± 0.06| 0.09 ± 0.05 |
> > > > |User Level Data - II| 0.31 ± 0.25 | 0.27 ± 0.22| 0.18 ± 0.18| 0.12 ± 0.19| 0.07 ± 0.15| 0.00 ± 0.00|0.00 ± 0.00|
> > > >
> > > > **Stationarity Assumption:** We appreciate the reviewer's point about temporal dynamics in clickstream data. By Assumption 2, BLANCE assumes stationary causal structure for the current work. That said, we recognize the importance of non-stationary time series settings. We discuss these extensions in future work. Additionally, we would like to draw the reviewer's attention to the experiment setup in [1] Fujiwara et al 2023, where experiments are performed with simulation dataset where nodes have no semantic meaning, making it difficult to query the LM meaningfully for useful information.
> > > >
> > > > **Memorization Claim:** Just to clarify, we believe the point raised is not related to the memorization but to the ground-truth causal structure creation in Google user level datasets. There seems to be a misunderstanding which merits clarification. The Google data from Kaggle is used to obtain the causal structure using DirectLiNGAM. Then we simulate data from this causal structure, considering this as the ground truth. Thus, we do not work with the data that resulted in the causal structure. The causal structure is held as a ground truth, and data simulated from it is used by our algorithm.
> > > > This clarification should allay concerns related to inductive biases or estimated graph being systematically wrong. We discuss this in Appendix Section B of the manuscript.

---

### Official Review · Reviewer_L954 · 2025-11-07

**Soundness:** 2
**Presentation:** 3
**Contribution:** 2
**Rating:** 6
**Confidence:** 4

**Summary:**

The paper presents BLANCE, a framework that treats large language models (LLMs) as noisy experts for causal discovery in Partial Ancestral Graphs (PAGs). BLANCE models large language models (LLMs) as stochastic experts answering edge-level queries.
Given sequential batches of observational data, an initial PAG from FCI is refined via an information-gain criterion that selects a small subset of edges to query. The LLM responses are aggregated into histograms, used to compute a dynamic inclusion threshold that updates priors for the next round. Additionally, BLANCE uses an EM-style parameter-estimation step where the LLM suggests priors over latent confounders. Experiments on six standard small-graph benchmarks (5–37 variables) show consistent improvements in Modified SHD and SID over FCI variants and prior LM-assisted baselines.

**Strengths:**

1) Representation shift: DAG → PAG lets the LLM return “uncertain” edges (◦,→, ↔) instead of forcing a directed edge.
2) The histogram → threshold → prior loop is intuitive and ablated effectively.
3) Sequential optimisation balances exploration (entropy) vs. exploitation (proximity to inclusion threshold).
4) Joint structure + parameter pipeline: LLM also proposes Gaussians for latent confounders (Red-Wine example).
5) Model-agnostic: works with any PAG-producing algorithm (FCI, RFCI, GFCI).
6) Clear motivation and algorithmic flow; appendix includes prompts and dataset details.

**Weaknesses:**

1) The sequential setting invites a bandit-style analysis, but no finite-sample or regret guarantees are provided.
2) All structure results are synthetic; the only real dataset (Red Wine Quality) is used for parameter estimation, not structural recovery.
3) LLM usage cost (API calls, tokens, wall time) is unreported, limiting reproducibility of practical feasibility.
4) Missing connection and baseline of amortized expert-aided discovery [1] similarly integrates expert feedback (human and LLM) in sequential fashion (with information-gain type criteria) iand handles latent confounders via learned proposal distributions. A short conceptual or empirical comparison would clarify relative cons-pros.
5) LLM-induced bias uncalibrated – temperature ablation (Table A3) shows modest effect; no mapping from LM log-prob to posterior weight.

**Questions:**

1) What is the wall-clock time and token usage for the largest benchmark?
2) How does the selection policy scale to d ≈ 50-100 variables?
3) Is it better to use the LLM-reported prob of edges (in text form) or log-prob induced by embedding? Could LM probabilities be calibrated into quantitative posteriors instead of binary votes?
4) What is the performance in real-world causal graphs (e.g., Sachs, or expert-labeled biomedical/economic data)?
5) Would a generative or amortized formulation (e.g., GFlowNet-style ancestral sampling like [1]) reduce query cost or improve uncertainty estimation? How does your method compared with such methods?
6) Any theoretical path toward regret or error-rate bounds under an idealized consistent expert?

[1] Expert-Aided Causal Discovery of Ancestral Graphs https://arxiv.org/pdf/2309.12032

---

> ### Author Response · Authors · 2025-11-21
>
> We thank the reviewer for their thoughtful comments. We appreciate the strengths of our work that you have noted. We address both questions and weaknesses below. Recall BLANCE is our method.
>
> **(W1 & Q6) Bandit-style analysis:** We provide the connection with stochastic multi-arm bandit and the regret interpretation in the Appendix of the manuscript, **Section E**. We discuss in Section E2 and E3 about the challenges in deriving the Multi-arm bandit regret bounds and pave the path for future work involving contextual and Bayesian bandits that incorporate evolving priors, combinatorial bandits for structured edge dependencies, and information-theoretic regret bounds based on entropy reduction.
>
> **(W2 & Q4) Real-world datasets:** Our evaluation includes six datasets: *Asia, Earthquake, User Level Data-I, User Level Data-II, Child, and Alarm*. For *Asia, Earthquake, Child, Alarm*, the causal structure is expert-guided corresponding to real-world scenarios. This affords a ground truth causal structure, enabling evaluation with structural accuracy metrics such as Modified SHD, SID, Precision, and Recall. The *User Level Data-I* and *User Level Data-II* datasets are drawn from **real-world** datasets sourced from Kaggle, reflecting user behavior in actual ecommerce, web setting (public dataset link provided in paper, also reproducing here, https://www.kaggle.com/datasets/bigquery/google-analytics-sample/data). However, these datasets do not come with known causal graphs. To enable evaluation on them, we follow a common practice: we run a causal discovery algorithm (DirectLiNGAM) on the dataset to generate a true causal graph. This inferred structure then forms a ground truth, allowing us to compute the evaluation metrics. Given the absence of ground truth causal graphs and the need for evaluation, our approach is principled and follows in the footsteps of significant prior research.
>
> **(W3 & Q1) LLM usage cost** As requested, the average monetary cost (in USD) using the GPT-3.5-Turbo model on the Earthquake dataset is shown below:
>
> | Method | Avg. Cost ($) | Mod. SHD |
> | -------- | -------- | ---- |
> | Triplet Prompting (DAG)     | 0.0356    | $1.4 \pm 1.4$ |
> | BLANCE     | 0.0304 | $1.0 \pm 0.6$ |
>
> Notably, triplet prompting consider just a single sample from the LM and is not grounded on the observational data. BLANCE is explicitly designed to operate under tight computational or monetary budgets. It supports a fixed query budget, and within this constraint, it uses an adaptive strategy to prioritize only the most uncertain or informative edges for refinement. This ensures that even when resources are limited, BLANCE allocates LM calls strategically to maximize structural improvements attributed to **sequential optimization**. As a result, BLANCE achieves optimal accuracy given the fixed budget.
>
> **(W4 & Q5) Comparison with Expert-Aided Causal Discovery of Ancestral Graphs[1]:** Thank you for bringing this paper to our attention. We were not aware of it. The arXiv version in your link is dated October 10, 2025, **after** the ICLR submission deadline. That said, we read the paper now and in our future work will refer to it. One important distinction in Causal Structure Discovery is that while our method is anchored in sequential batched data, theirs is developed for a setup where the full dataset is available (e.g. Sachs data they use). Other differences include: the updating they perform is around information gain, and they do not perform edge-weight estimation. Moreover, we did not find a publicly available code in this paper for us to quickly implement during rebuttal, in datasets we use and evaluate with the same metrics as in our paper.
>
> **(W5 & Q3) Use LM log-probability to get quantitative posteriors:** We agree this is intuitively appealing. However, during inference, the language model log-probability are over tokens, that are not well-calibrated and it does not correspond to the absolute certainty in a response [1]. To overcome this obstacle, we model LM responses as noisy samples aggregated over multiple calls, which allows empirical estimation of uncertainty through histogram-based posteriors. This repeated sampling approach empirically reduces hallucination effects and captures the LM’s evolving beliefs more robustly than a single-shot uncertainty estimate.
>
> [1] [Language Model Probabilities are Not Calibrated in Numeric Contexts](https://aclanthology.org/2025.acl-long.1417/) (Lovering et al., ACL 2025)

---

> > ### Author Response · Authors · 2025-11-21
> >
> > **(Q2) Selection policy scaling to larger graphs:** The selection policy involves calculating the selection score (*cf.* Eq. 9) for all candidate edges identified by the causal discovery algorithm. Importantly, (1) the selection score for each edge is independent and can be computed in parallel, and (2) since BLANCE uses a data-first approach, the FCI algorithm substantially prunes the search space of candidate edges by identifying conditional independence relationships from the data. Therefore, the actual number of edges requiring LM queries is typically much smaller than $O(d^2)$.
> >
> > We have updated the manuscript with new LM model experiments (Table 5) as suggested by reviewers. We have updated the future work section. Please let us know if you have any further questions or suggestions.

---

### Author Response · Authors · 2025-11-27

Dear reviewers,

Once again, we thank you very much for the thoughtful reviews on our manuscript.

With the discussion period deadline approaching fast, we will appreciate hearing your response to the individual rebuttal. We hope our rebuttals have addressed your concerns. If you seek further clarification on any point, we will be happy to respond.

We greatly appreciate your time and effort.

---

### Author Response · Authors · 2025-12-02
**Final Remarks**

We take this opportunity to thank the reviewers for their thoughtful comments, which have helped in sharpening the results and in improving the manuscript overall. First, we summarize the previous discussions. Then, we add a briefing of deliberations for each reviewer and share the updates. The updates have been incorporated in the revised manuscript, which is available on the portal.

**Summary:** We offer a new framework to perform causal structure learning in a hybrid setting of sequential, batch-wise data and LM. Sequential, batched data is pervasive in online firms' web data, but is less attended in causal research. Crucially, we also allow for selection bias in this type of data *and* for the LM's response to be noisy, both affording novelty and generality. We propose a shift from DAG to PAG to represent uncertainty in edge orientation, to properly reflect our setting. Our causal structure discovery itself is Bayesian, where we inject additional novelty in using a sequential optimization method for selecting maximally informative LM edge queries under a budget constraint. Morever, we extend structure learning to estimation of edge-weights. The parameter estimation is Bayesian too and robustly integrates noisy LM priors with batched data. Across a host of SOTA baselines we show that our framework outperforms them on several metrics in a statistically significant manner ($ \alpha = 0.05$).

Across reviews, there is an agreement that the paper addresses an important and underexplored setting of causal discovery with sequential, potentially biased batches of data and LM-bias. Reviewers *VGuz* and *nVWr* highlight the novelty and clarity of the Bayesian integration of LMs as noisy experts, noting the well-motivated hybrid framework and the coherent structure of the method. Reviewers *L954* and *yAbn* emphasize the shift from DAGs to PAGs as a key strength enabling uncertainty modeling in edge orientations. Multiple reviewers *(VGuz, L954, yAbn)* appreciate the sequential optimization strategy that efficiently selects informative LLM queries, reducing inference cost while improving accuracy. Strong reproducibility, comprehensive experimental documentation, and clear appendices were also positively noted *(VGuz, nVWr)*, alongside the clean algorithmic flow and consistent empirical improvements over baselines *(nVWr, L954)*.

At the same time, each reviewer raised concerns and questions. In pointwise rebuttal to each reviewer and in response thereupon in the discussion phase, we addressed in detail each question and weakness. As asked by reviewers, we shared new experimental results that included additional SOTA baselines, and recent and different families of LM. Those results, available in the revised manuscript, continue to favor our BLANCE framework.

**Manuscript Updates During Discussion:** During the rebuttal and discussion period, we made some really important updates to the manuscript which, we think, addresses the issues raised by the reviewers. The detailed discussions with reviewers are available in the thread below. Here, we highlight the major ones.

1. We experiment with two more recent and different LM families, **Llama3** and **Qwen3**. They showcase the superior performance of BLANCE across different model families (GPT, Llama3, Qwen3), architectures and memorization profiles. These were added to the manuscript (Table 5).
2. We experiment with a recently proposed approach *Efficient Causal Graph Discovery Using Large Language Models Jiralerspong et al. 2024 (BLS)* using **GPT4.1-nano (2025-04-14)** and showcase the superior performance of BLANCE, and also showcase robustness and independence over LM memorization. These were also added to the manuscript (Table 5).
3. We added a discussion on **memorization**, the issue raised by Reviewer *yAbn* and *nVWr*. The discussion is added to the manuscript (Section 5.3)
5. Reviewer *yAbn* agrees that F1 is a proxy for ambiguity compression. However, wondered if there is a method to quantify it. We discussed the metrics **Mod SHD** and **SID** that are specifically chosen to quantify this uncertainty in the causal structure and performed an additional experiment as a response to *yAbn*. We reported **mean histogram entropy** over batches, which decreases, showcasing that ambiguity compression occurs. We have added this result to the manuscript (Table A4).
6. We have added an expanded section on **Limitations and Future Work** in the manuscript (Section 6), as asked by the reviewers.
7. If we may refer to the final review comment from *yAbn*.\
*"Please refer the weaknesses section, and clarify the points. I am happy to increase my score if the authors can provide justification and empirical results to tackle these points, especially the memorization issue."*

In conclusion, we appreciate all the reviewers' suggestions and points raised. We believe we were able to incorporate them and address the points raised via new experiments and discussion.

---

### Meta-Review · Area_Chair_K53p · 2026-01-05

**Summary:**

This paper presents develops a method for causal discovery where data arrives in sequential, batch-wise fashion. The method involves an EM-style parameter-estimation step where the LLM suggests priors over latent confounder. While the paper contains valuable contributions, the reviewers pointed out significant issues including requiring restrictive assumptions, providing no theoretical guarantee, and lacks sufficient novelty in the methodology. Given the highly competitive nature of this conference, I recommend rejection. However, I encourage the authors to address these feedbacks to strengthen their work for a future submission.

**Reviewer Concerns:**

Concerns addressed during rebuttal:
- LLM usage cost (API calls, tokens, wall time) is unreported
- Lack of evaluation for each moving part of the method
- Lack of empirical studies for showing generalization of the proposed framework to other causal discovery methods
- Motivation about sequential batch setting
- Background knowledge of LLM might contradict a given dataset domain
- The final metrics focus only on SHD or final graph based edges, and there is no analysis of ambiguity or how LLM induced bias is reduced by the proposed method.
- Reviewers pointed out the memorization issues by LLMs especially for bnlearn dataset and the lack of recent real world datasets.

Concerns that are not fully addressed:
- The methodology lacks sufficient novelty. Different components of the proposed method, including PAG estimation and Bayesian approach, are not new.
- The method requires restrictive assumptions (including linear Gaussian SEM for Bayesian parameter estimation, stationary data, and single confounder), thereby restricting the method's applicability.
- The method relies on the LLM's knowledge and is itself heuristic, providing no theoretical guarantee.
- The graphs are relatively small, which makes guessing easier.

**Reviewer Scores:**

Reviewer yAbn may have changed their score to 4 if they had been able to participate fully in the discussion.

---

### Decision · Program_Chairs · 2026-01-26

Reject